# Unit Ball Model for Embedding Hierarchical Structures in the Complex Hyperbolic Space

## Abstract

Learning the representation of data with hierarchical structures in the hyperbolic space attracts increasing attention in recent years. Due to the constant negative curvature, the hyperbolic space resembles tree metrics and captures the tree-like properties naturally, which enables the hyperbolic embeddings to improve over traditional Euclidean models. However, most real-world hierarchically structured data such as taxonomies and multitree networks have varying local structures and they are not trees, thus they do not ubiquitously match the constant curvature property of the hyperbolic space. To address this limitation of hyperbolic embeddings, we explore the complex hyperbolic space, which has the variable negative curvature, for representation learning. Specifically, we propose to learn the embeddings of hierarchically structured data in the unit ball model of the complex hyperbolic space. The unit ball model based embeddings have a more powerful representation capacity to capture a variety of hierarchical structures. Through experiments on synthetic and real-world data, we show that our approach improves over the hyperbolic embedding models significantly. We also explore the competence of complex hyperbolic geometry on the multitree structure and $1$-$N$ structure.

## 1 Introduction

Representation learning of data with hierarchical structures is an important machine learning task with many applications, such as taxonomy induction (Fu et al., 2014) and hypernymy detection (Shwartz et al., 2016). In recent years, the hyperbolic embeddings (Nickel and Kiela, 2017; 2018) have been proposed to improve the traditional Euclidean embedding models. The constant negative curvature of the hyperbolic space produces a manifestation that the hyperbolic space can be regarded as a continuous approximation to trees (Krioukov et al., 2010). The hyperbolic space is capable of embedding any finite tree while preserving the distances approximately (Gromov, 1987).

However, most real-world hierarchical data do not belong to tree structures since they can have varying local structures while being tree-like globally. For example, the taxonomies such as WordNet (Miller, 1995) and YAGO (Suchanek et al., 2007) contain many $1$-$N$ (1 child links to multiple parents) cases and multitree structures (Griggs et al., 2012), which are much more complicated than the tree structure. In consequence, the hyperbolic space which resembles tree metrics has limitations on capturing the general hierarchically structured data.

To address the challenge, in this paper, we propose a new approach to learning the embeddings of hierarchically structured data. Specifically, we embed the hierarchical data into the unit ball model of the complex hyperbolic space. The unit ball model is a projective geometry based model to identify the complex hyperbolic space. One of the main differences between the complex and the real hyperbolic space is that the curvature is non-constant in the complex hyperbolic space. In practice, the variable negative curvature makes the complex hyperbolic space more flexible in handling varying structures while the tree-like properties are still retained.

For empirical evaluation, we evaluate different geometrical embedding models on various hierarchically structured data, including synthetic graphs and real-world data. The experimental results demonstrate the advantages of our approach. In addition, we investigate two specific structures in which complex hyperbolic geometry shows outstanding performances, namely the multitree structure and $1$-$N$ structure, which are highly common and typical in real-world data. To summarize, our work has the following main contributions: 1. We present a novel embedding approach based on the

complex hyperbolic geometry to handle data with complicated and various hierarchical structures. To the best of our knowledge, our work is the first to propose complex hyperbolic embeddings. 2. We introduce the embedding algorithm in the unit ball model of the complex hyperbolic space. We formulate the learning and Riemannian optimization in the unit ball model. 3. We evaluate our approach with experiments on an extensive range of synthetic and real-world data and show the remarkable improvements of our approach.

## 2 RELATED WORK

**Hyperbolic embeddings.** Hyperbolic embedding methods have become the leading approach for representation learning of hierarchical structures. Nickel and Kiela (2017) learned the representations of hierarchical graphs in the Poincaré ball model of the hyperbolic space and outperformed the Euclidean embedding methods for taxonomies. The Poincaré embedding model was then improved by follow-up works on hyperbolic emebddings (Ganea et al., 2018a; Nickel and Kiela, 2018). These methods learned the hyperbolic embeddings by Riemannian optimization (Bonnabel, 2013), which was further improved by the Riemannian adaptive optimization (Bécigneul and Ganea, 2019).

Another branch of study (Sala et al., 2018; Sonthalia and Gilbert, 2020) learned the hyperbolic embeddings through combinatorial construction. Instead of optimizing the soft-ranking loss by Riemannian optimization as in (Nickel and Kiela, 2017; 2018), the construction-based methods minimize the reconstruction distortion by combinatorial construction. However, both the optimization-based and construction-based hyperbolic embeddings suffer from the limitation in hierarchical graphs with varying local structures. To tackle the challenge, Gu et al. (2019) extended the construction-based method by jointly learning the curvature and the embeddings of data in a product manifold. Although it can provide a better representation than a single space with constant curvature, it is impractical to search for the best manifold combination among enormous combinations for each new structure.

Motivated by the promising results of previous works, extensions to the multi-relational graph hyperbolic embeddings (Balazevic et al., 2019; Chami et al., 2020; Sun et al., 2020) and hyperbolic neural networks (Ganea et al., 2018b; Gülçehre et al., 2019; Liu et al., 2019; Chami et al., 2019; Zhu et al., 2020; Dai et al., 2021a; Shimizu et al., 2021) were explored. Notably, (Chami et al., 2019; 2020) leverages the trainable curvature to compensate for the disparity between the actual data structures and the constant-curvature hyperbolic space, where each layer in the graph neural network or each relation in the multi-relational graph has its own curvature parameterization. Since we only focus on the single-relation graph embeddings and taxonomy embeddings in this work, we do not evaluate the multi-relational knowledge graph embedding models or the neural networks in our tasks.

The hyperbolic learning also inspired other research tasks and applications, such as classification (Cho et al., 2019), image reconstruction (Skopek et al., 2020), text generation (Dai et al., 2021b), etc.

**Complex embeddings.** The traditional knowledge graph embeddings were learned in the real Euclidean space (Nickel et al., 2011; Bordes et al., 2013; Yang et al., 2015) and were used for knowledge graph inference and reasoning. In recent years, several works suggested utilizing the complex Euclidean space for inferring more relation patterns, such as ComplEx (Trouillon et al., 2016) and RotatE (Sun et al., 2019). These models have been demonstrated to be effective in knowledge graph embeddings. The success of the complex embeddings reveals the potential of the complex space and inspires us to explore the complex hyperbolic space.

## 3 PRELIMINARIES

### 3.1 HYPERBOLIC GEOMETRY

Hyperbolic space[1] is a homogeneous space with constant negative curvature.[2] In the hyperbolic space $\mathbb{H}^n_{\mathbb{R}}(K)$ of dimension $n$ and curvature $K$, the volume of a ball grows exponentially with its radius $\rho$:

$$vol(B_{\mathbb{H}^n_{\mathbb{R}}(K)}(\rho)) \sim e^{\sqrt{-K}(n-1)\rho}. \tag{1}$$

---

[1]In this paper, we use *hyperbolic space* to refer to real hyperbolic space and *hyperbolic embeddings* to refer to real hyperbolic embeddings for avoiding wordiness.

[2]In this paper, *curvature* refers to the *sectional curvature*. Please see Appendix A.1 for definition.

Contrastively, in the Euclidean space $\mathbb{E}^n$, the curvature is $0$ and the volume of a ball grows polynomially with its radius:

$$vol(B_{\mathbb{E}^n}(\rho)) = \frac{\pi^{n/2}}{\Gamma(n/2)}\rho^n \sim \rho^n. \tag{2}$$

The exponential volume growth rate enables the hyperbolic space to have powerful representation capability for tree structures since the number of nodes grows exponentially with the depth in a tree, while the Euclidean space is too flat and narrow to embed trees.

## 3.2 COMPLEX HYPERBOLIC GEOMETRY

Complex hyperbolic space is a homogeneous space of variable negative curvature. Its ambient Hermitian vector space $\mathbb{C}^{n,1}$ is the complex Euclidean space $\mathbb{C}^{n+1}$ endowed with some Hermitian form $\langle\!\langle \mathbf{z}, \mathbf{w} \rangle\!\rangle$, where $\mathbf{z}, \mathbf{w} \in \mathbb{C}^{n+1}$. Then the Hermitian space $\mathbb{C}^{n,1}$ can be divided into three subsets: $V_- = \{\mathbf{z} \in \mathbb{C}^{n,1} | \langle\!\langle \mathbf{z}, \mathbf{z} \rangle\!\rangle < 0\}$, $V_0 = \{\mathbf{z} \in \mathbb{C}^{n,1} - \{\mathbf{0}\} | \langle\!\langle \mathbf{z}, \mathbf{z} \rangle\!\rangle = 0\}$, and $V_+ = \{\mathbf{z} \in \mathbb{C}^{n,1} | \langle\!\langle \mathbf{z}, \mathbf{z} \rangle\!\rangle > 0\}$. Let $\mathbb{P}$ be a projection map $\mathbb{P} : \mathbb{C}^{n,1} - \{z_{n+1} = 0\} \to \mathbb{C}^n$, i.e.,

$$\mathbb{P} : \begin{bmatrix} z_1 \\ \vdots \\ z_{n+1} \end{bmatrix} \mapsto \begin{bmatrix} z_1/z_{n+1} \\ \vdots \\ z_n/z_{n+1} \end{bmatrix}, \text{where } z_{n+1} \neq 0. \tag{3}$$

Then the complex hyperbolic space $\mathbb{H}_{\mathbb{C}}^n$ and its boundary $\partial\mathbb{H}_{\mathbb{C}}^n$ are defined using the projectivization:

$$\mathbb{H}_{\mathbb{C}}^n = \mathbb{P}V_-, \qquad \partial\mathbb{H}_{\mathbb{C}}^n = \mathbb{P}V_0. \tag{4}$$

The curvature of the complex hyperbolic space is summarized by (Goldman, 1999) as follows:

**Theorem 1.** *The curvature is not constant in $\mathbb{H}_{\mathbb{C}}^n$. It is pinched between $-1$ (in the directions of complex projective lines) and $-1/4$ (in the directions of totally real planes).*

We leave the full proof in Appendix B. The non-constant curvature, which we expect to be favorable for embedding various hierarchical structures, is one of the main differences between $\mathbb{H}_{\mathbb{C}}^n$ and $\mathbb{H}_{\mathbb{R}}^n$.

The complex hyperbolic space also has the tree-like exponential volume growth property. The volume of a ball with radius $\rho$ in $\mathbb{H}_{\mathbb{C}}^n$ is given by

$$vol(B_{\mathbb{H}_{\mathbb{C}}^n}(\rho)) = \frac{8^n \sigma_{2n-1}}{2n}\sinh^{2n}(\rho/2) \sim e^{n\rho}, \tag{5}$$

where $\sigma_{2n-1} = 2\pi^n/n!$ is the Euclidean volume of the unit sphere $S^{2n-1} \in \mathbb{C}^n$.

From the properties of the complex hyperbolic geometry, we expect that the complex hyperbolic space can naturally handle data with diverse local structures in virtue of the variable curvature as presented in Theorem 1 while preserving the tree-like properties as shown in Eq. (5).

From this section, we see that complex hyperbolic geometry and hyperbolic geometry are typically of different characteristics. The $n$-dimensional ($n$-d) complex hyperbolic space is not simply the $2n$-d hyperbolic space or the product of two $n$-d hyperbolic spaces. This implies that our complex hyperbolic embedding model is intrinsically different from the hyperbolic embedding methods (Nickel and Kiela, 2017; 2018) or the product manifold embeddings (Gu et al., 2019).

## 4 UNIT BALL EMBEDDINGS

We propose to embed the hierarchically structured data into the unit ball model of the complex hyperbolic space. In this section, we introduce our approach in detail.

### 4.1 THE UNIT BALL MODEL

The unit ball model is one model used to identify the complex hyperbolic space, which can be derived via the projective geometry (Goldman, 1999). We now provide the necessary derivation.

Take the Hermitian form of $\mathbb{C}^{n,1}$ in Section 3.2 to be a standard Hermitian form:

$$\langle\!\langle \mathbf{z}, \mathbf{w} \rangle\!\rangle = z_1 \overline{w_1} + \cdots + z_n \overline{w_n} - z_{n+1} \overline{w_{n+1}}, \tag{6}$$

where $\overline{w}$ is the conjugate of $w$. Take $z_{n+1} = 1$ in the projection map $\mathbb{P}$ in Eq. (3). Then from Eq. (4), we can derive the formula of the unit ball model:

$$\mathcal{B}_{\mathbb{C}}^n = \mathbb{P}(\{\mathbf{z} \in \mathbb{C}^{n,1} | \langle\!\langle \mathbf{z}, \mathbf{z} \rangle\!\rangle < 0\}) = \{(z_1, \cdots, z_n, 1) | |z_1|^2 + \cdots + |z_n|^2 < 1\}. \tag{7}$$

The metric on $\mathcal{B}_{\mathbb{C}}^n$ is Bergman metric, which takes the formula below in 2-d case:

$$ds^2 = \frac{-4}{\langle\!\langle \mathbf{z}, \mathbf{z} \rangle\!\rangle^2} \det \begin{bmatrix} \langle\!\langle \mathbf{z}, \mathbf{z} \rangle\!\rangle & \langle\!\langle d\mathbf{z}, \mathbf{z} \rangle\!\rangle \\ \langle\!\langle \mathbf{z}, d\mathbf{z} \rangle\!\rangle & \langle\!\langle d\mathbf{z}, d\mathbf{z} \rangle\!\rangle \end{bmatrix}. \tag{8}$$

The distance function on $\mathcal{B}_{\mathbb{C}}^n$ is given by

$$d_{\mathcal{B}_{\mathbb{C}}^n}(\mathbf{z}, \mathbf{w}) = arcosh\Big(2\frac{\langle\!\langle \mathbf{z}, \mathbf{w} \rangle\!\rangle \langle\!\langle \mathbf{w}, \mathbf{z} \rangle\!\rangle}{\langle\!\langle \mathbf{z}, \mathbf{z} \rangle\!\rangle \langle\!\langle \mathbf{w}, \mathbf{w} \rangle\!\rangle} - 1\Big), \tag{9}$$

Note that there are other choices of the Hermitian form $\langle\!\langle \mathbf{z}, \mathbf{w} \rangle\!\rangle$, which corresponds to other models of complex hyperbolic geometry, such as the Siegel domain model. We choose the unit ball model for the relatively simple formula as well as convenient computations of the metric and distance function.

### 4.2 Embeddings in the Unit Ball Model

Given the hierarchical data containing a set of nodes $X = \{x_p\}_{p=1}^m$ and a set of edges $E = \{(x_p, x_q) | x_p, x_q \in X\}$, we aim to learn the embeddings of the nodes $\mathbf{Z} = \{\mathbf{z}_p\}_{p=1}^m$, where $\mathbf{z}_p \in \mathcal{B}_{\mathbb{C}}^n$.

The objective of the embeddings is to recover the structures of input data, including the distances between the nodes as well as the partial order in the hierarchies. Here we adopt the soft ranking loss used in the Poincaré ball embeddings (Nickel and Kiela, 2017) and the hyperboloid embeddings (Nickel and Kiela, 2018), which aims at preserving the hierarchical relationships among nodes:

$$L = \sum_{(x_p, x_q) \in E} \log \frac{e^{-d_{\mathcal{B}_{\mathbb{C}}^n}(\mathbf{z}_p, \mathbf{z}_q)}}{\sum_{x_k \in \mathcal{N}(x_p)} e^{-d_{\mathcal{B}_{\mathbb{C}}^n}(\mathbf{z}_p, \mathbf{z}_k)}}, \tag{10}$$

where $\mathcal{N}(x_p) = \{x_k : (x_p, x_k) \notin E\} \cup \{x_p\}$ is the set of negative examples for $x_p$ together with $x_p$. $d_{\mathcal{B}_{\mathbb{C}}^n}$ is the distance function in the unit ball model given in Eq. (9). The minimization of $L$ makes the connected nodes closer in the embedding space than those with no observed edges.

The learning process implicitly aligns the geometric structures of the embedding space and the underlying graph structures of data since the loss function aims at preserving the hierarchical relationships among nodes while the underlying graph structures are reflected by the hierarchical relationships. We learn the embeddings in the unit ball model, where the variable negative curvature of the complex hyperbolic space provides the capacity to deal with more varying structures. The experiments in Section 5 exhibit that the unit ball model learns the high-quality embeddings and captures the various hierarchical structures.

### 4.3 Riemannian Optimization in the Unit Ball Model

We learn the embeddings $\mathbf{Z} = \{\mathbf{z}_p\}_{p=1}^m$ through solving the optimization problem with constraint:

$$\mathbf{Z} \leftarrow \arg\min_{\mathbf{Z}} L \qquad s.t. \forall \mathbf{z}_p \in \mathbf{Z}, \mathbf{z}_p \in \mathcal{B}_{\mathbb{C}}^n. \tag{11}$$

For the optimization problems in Riemannian manifolds, Bonnabel (2013) presented the Riemannian stochastic gradient descent (RSGD) algorithm, which we employ to optimize Eq. (11). To update an embedding $\mathbf{z} \in \mathcal{B}_{\mathbb{C}}^n$,[3] we need to obtain its Riemannian gradient $\nabla_R$. Specifically, the embedding is updated at the $t$-th iteration by $\mathbf{z}^{(t)} \leftarrow \mathbf{z}^{(t-1)} - \eta^{(t)} \nabla_R L(\mathbf{z})$, where $\eta^{(t)}$ is the learning rate at the $t$-th iteration and $\nabla_R L(\mathbf{z})$ is the Riemannian gradient of $L(\mathbf{z})$.

---

[3]Here we omit the subscript of $\mathbf{z}_p$ for concision.

---

**Algorithm 1** RSGD of the unit ball embeddings.

---

**Input:** initialization $\mathbf{z}^{(0)}$, number of iterations $T$, learning rates $\{\eta^{(t)}\}_{t=1}^{T}$.
**for** $t = 1$ **to** $T$ **do**
    Compute $\frac{\partial d_{\mathcal{B}_{\mathbb{C}}^{n}}}{\partial \mathbf{x}}$ and $\frac{\partial d_{\mathcal{B}_{\mathbb{C}}^{n}}}{\partial \mathbf{y}}$ by Eqs. (13) and (14).
    Compute $\nabla_E L(\mathbf{z})$ and $\nabla_R L(\mathbf{z})$ by Eq. (12).
    Update $\mathbf{z}^{(t)}$ by Eq. (16).
**end for**

---

The Riemannian gradient $\nabla_R$ can be derived from rescaling the Euclidean gradient $\nabla_E$ with the inverse of the metric tensor $ds^2$ in Eq. (8). Apply the chain rule of differential functions and we have:

$$\nabla_R L(\mathbf{z}) = \frac{1}{ds^2} \nabla_E L(\mathbf{z}) = \frac{1}{ds^2} \frac{\partial L(\mathbf{z})}{\partial d_{\mathcal{B}_{\mathbb{C}}^{n}}(\mathbf{z}, \mathbf{w})} \nabla_E d_{\mathcal{B}_{\mathbb{C}}^{n}}(\mathbf{z}, \mathbf{w}). \tag{12}$$

$\frac{\partial L(\mathbf{z})}{\partial d_{\mathcal{B}_{\mathbb{C}}^{n}}(\mathbf{z},\mathbf{w})}$ is trivial to compute from Eq. (10). In practical training, we implement and compute the complex hyperbolic embedding as its real part and imaginary part, i.e., $\mathbf{z} = \mathbf{x} + i\mathbf{y}$, where $i$ represents the *imaginary unit*, i.e., $i^2 = -1$. In order to get the gradient of the distance function $\nabla_E d_{\mathcal{B}_{\mathbb{C}}^{n}}(\mathbf{z}, \mathbf{w})$ in Eq. (12), we get the partial derivative with regard to the real part and the imaginary part, i.e., $\nabla_E d_{\mathcal{B}_{\mathbb{C}}^{n}}(\mathbf{z}, \mathbf{w}) = \frac{\partial d_{\mathcal{B}_{\mathbb{C}}^{n}}(\mathbf{z},\mathbf{w})}{\partial \mathbf{x}} + i\frac{\partial d_{\mathcal{B}_{\mathbb{C}}^{n}}(\mathbf{z},\mathbf{w})}{\partial \mathbf{y}}$.

The partial derivatives of the unit ball model distance take the following formulas:

$$\frac{\partial d_{\mathcal{B}_{\mathbb{C}}^{n}}}{\partial \mathbf{x}} = \frac{4}{\sqrt{p^2 - 1}} \left( \frac{Re(\langle\!\langle \mathbf{z}, \mathbf{w} \rangle\!\rangle \mathbf{w})}{\langle\!\langle \mathbf{z}, \mathbf{z} \rangle\!\rangle \langle\!\langle \mathbf{w}, \mathbf{w} \rangle\!\rangle} - \frac{\langle\!\langle \mathbf{z}, \mathbf{w} \rangle\!\rangle \langle\!\langle \mathbf{w}, \mathbf{z} \rangle\!\rangle \mathbf{x}}{\langle\!\langle \mathbf{z}, \mathbf{z} \rangle\!\rangle^2 \langle\!\langle \mathbf{w}, \mathbf{w} \rangle\!\rangle} \right), \tag{13}$$

$$\frac{\partial d_{\mathcal{B}_{\mathbb{C}}^{n}}}{\partial \mathbf{y}} = \frac{4}{\sqrt{p^2 - 1}} \left( \frac{Im(\langle\!\langle \mathbf{z}, \mathbf{w} \rangle\!\rangle \mathbf{w})}{\langle\!\langle \mathbf{z}, \mathbf{z} \rangle\!\rangle \langle\!\langle \mathbf{w}, \mathbf{w} \rangle\!\rangle} - \frac{\langle\!\langle \mathbf{z}, \mathbf{w} \rangle\!\rangle \langle\!\langle \mathbf{w}, \mathbf{z} \rangle\!\rangle \mathbf{y}}{\langle\!\langle \mathbf{z}, \mathbf{z} \rangle\!\rangle^2 \langle\!\langle \mathbf{w}, \mathbf{w} \rangle\!\rangle} \right), \tag{14}$$

where $p = \cosh(d_{\mathcal{B}_{\mathbb{C}}^{n}}(\mathbf{z}, \mathbf{w}))$. $Re(\cdot)$ and $Im(\cdot)$ denote the real and the imaginary part respectively. The full derivation of Eqs. (13) and (14) is given in Appendix C.

Since the embedding $\mathbf{z}$ should be constrained within the unit ball model, we apply the same projection strategy as (Nickel and Kiela, 2017) via a small constant $\varepsilon$:

$$proj(\mathbf{z}) = \mathbf{z}/(|\mathbf{z}| - \varepsilon), \text{ if } |\mathbf{z}| \geq 1, \text{ else } \mathbf{z}. \tag{15}$$

To sum up, the update of $\mathbf{z}$ at the $t$-th iteration is

$$\mathbf{z}^{(t)} \leftarrow proj\big(\mathbf{z}^{(t-1)} - \eta^{(t)} \nabla_R L(\mathbf{z})\big) = proj\big(\mathbf{z}^{(t-1)} - \eta^{(t)} \frac{1}{ds^2} \nabla_E L(\mathbf{z})\big). \tag{16}$$

The RSGD steps of the unit ball embeddings are presented in Algorithm 1.

## 5 EXPERIMENTS

In experiments, we evaluate the performances of our approach and baselines on various hierarchical structures, including synthetic graphs and real-world data. We focus on the graph reconstruction task and the link prediction task. The main results are reported in this section. For more experiments, please refer to Appendix F.

### 5.1 EXPERIMENTAL SETTINGS

#### 5.1.1 DATA

We use synthetic and real-world data that exhibit underlying hierarchical structures to evaluate our approach. The public links or generator package links of the data are given in Appendix F.1.

**Synthetic.** We generate various balanced trees and compressed graphs using NetworkX package (Hagberg et al., 2008). For **balanced trees**, we generate the balanced tree with degree $r$ and depth $h$. For

Table 1: The real-world datasets statistics.

|  | Xiphophorus | ICD10 | YAGO3-wikiObjects | WordNet-noun |
|---|---|---|---|---|
| Nodes | 3,562 | 19,155 | 17,375 | 82,115 |
| Edges | 7,536 | 78,357 | 153,643 | 743,086 |
| Depth/ | 13 | 6 | 16 | 20 |
| Training edges | 7,536 | 70,521 | 138,277 | 668,776 |
| Valid/Test edges | 4,160 | 3,918 | 7,683 | 37,155 |
| $\delta$-hyperbolicity | 2.5 | 0.0 | 1.0 | 0.5 |

**compressed graphs**, we generate $k$ random trees on $m$ nodes and then aggregate their edges to form a graph. Some examples of the synthetic data are given in Appendix F.1.

**Xiphophorus.** The Xiphophorus is a multitree dataset from (Cui et al., 2013), which contains 160 trees representing mrbayes consensus trees inferred for different genomic regions on 26 Xiphophorus fishes. We use the saved MultiTree object in toytree package.

**ICD10.** The 10-th revision of International Statistical Classification of Diseases and Related Health Problems (ICD10) (Brämer, 1988) is a medical classification list provided by the World Health Organization. We construct its full transitive closure as the ICD10 dataset.

**YAGO3-wikiObjects.** YAGO3 (Mahdisoltani et al., 2015) is a huge semantic knowledge base. It provides a taxonomy derived from Wikipedia and WordNet. We extract the Wikipedia concepts and entities that are descendants of $\langle wikicat\_Objects \rangle$ as well as the hypernymy edges among them. We compute the transitive closure of the sampled taxonomy to construct the YAGO3-wikiObjects dataset.

**WordNet-noun.** WordNet (Miller, 1995) is a large lexical database. The hypernymy relation among all nouns forms a noun hierarchy. We use its full transitive closure as the WordNet-noun dataset.

For the multitree (Xiphophorus), we use the full dataset as the training set and the edges containing the leaf nodes as the test set. For each real-world taxonomy (ICD10, YAGO3-wikiObjects, WordNet-noun), we randomly split the edges into train-validation-test sets with the ratio $90\%:5\%:5\%$. We make sure that any node in the validation and test sets must occur in the training set since otherwise, it cannot be predicted. But the edges in the validation and test sets do not occur in the training set since they are disjoint. We provide the statistics of the real-world datasets in Table 1. The Gromov's $\delta$-hyperbolicity (Gromov, 1987) measures the tree-likeness of graphs (refer to Appendix D for definition). The lower $\delta$ corresponds to the more tree-like graph and trees have $0$ $\delta$-hyperbolicity.

### 5.1.2  TASKS

We evaluate the following two tasks:

**Graph reconstruction**: we train the embeddings of the full data and then reconstruct it from the embeddings. The task evaluates representation capacity. **Link prediction**: we train the embeddings on the training set and predict the edges in the test set. The task evaluates generalization performance.

### 5.1.3  BASELINES

We compare our approach **UnitBall** to the following methods: the sate-of-the-art combinatorial construction-based hyperbolic embedding method **TreeRep** (Sonthalia and Gilbert, 2020), the optimization-based hyperbolic embeddings in the **Poincaré** ball model (Nickel and Kiela, 2017) and the **Hyperboloid** model (Nickel and Kiela, 2018), the simple **Euclidean** embedding model. Euclidean, Poincaré, Hyperboloid, and our approach UnitBall use the same loss function but learn in the different geometrical spaces. Therefore, the comparisons reveal the capacities of different geometrical models in different spaces.

For the baselines, we use their released codes to train the embeddings. For all methods, the hyperparameters are tuned on each validation set for the link prediction task and on balanced tree-(15,3) for the graph reconstruction task. The hardware information is given in Appendix F.2 and the hyperparameters are listed in Appendix F.3. In all experiments, we report the mean results over 5 running executions. The code of our approach will be publicly available after the publication of the paper.

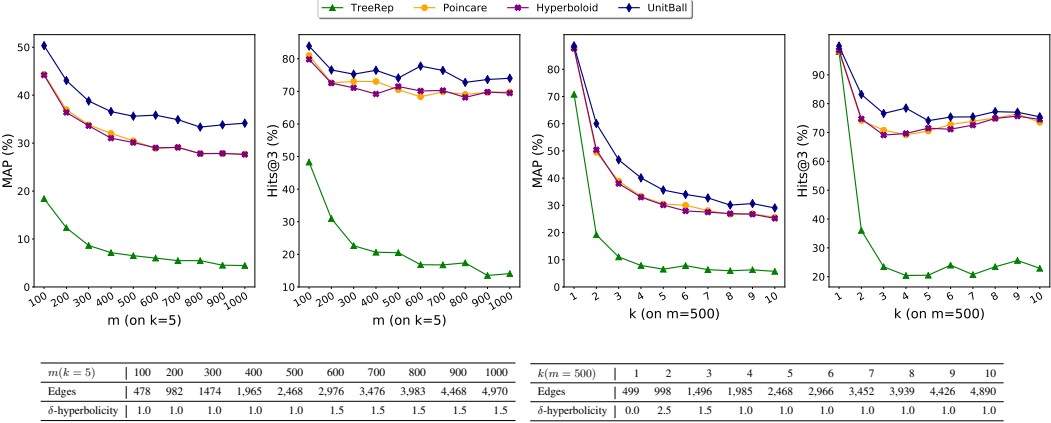

| $m(k=5)$ | 100 | 200 | 300 | 400 | 500 | 600 | 700 | 800 | 900 | 1000 |
|---|---|---|---|---|---|---|---|---|---|---|
| Edges | 478 | 982 | 1474 | 1,965 | 2,468 | 2,976 | 3,476 | 3,983 | 4,468 | 4,970 |
| $\delta$-hyperbolicity | 1.0 | 1.0 | 1.0 | 1.0 | 1.0 | 1.5 | 1.5 | 1.5 | 1.5 | 1.5 |

| $k(m=500)$ | 1 | 2 | 3 | 4 | 5 | 6 | 7 | 8 | 9 | 10 |
|---|---|---|---|---|---|---|---|---|---|---|
| Edges | 499 | 998 | 1,496 | 1,985 | 2,468 | 2,966 | 3,452 | 3,939 | 4,426 | 4,890 |
| $\delta$-hyperbolicity | 0.0 | 2.5 | 1.5 | 1.0 | 1.0 | 1.0 | 1.0 | 1.0 | 1.0 | 1.0 |

Figure 1: Evaluation of graph reconstruction on synthetic compressed graphs in 20-d embedding spaces (10-d complex hyperbolic space for UnitBall). $m$ represents the number of nodes in the graph while $k$ represents the number of random trees aggregated to the graph ($k$ controls the denseness and noise level of the graph). The statistics of the compressed graphs are provided in the tables.

### 5.1.4 EVALUATION

We use the mean average precision (**MAP**), mean reciprocal rank (**MRR**), and **Hits@N** as our evaluation metrics, which are widely used for evaluating ranking and link prediction. The details of prediction steps and the evaluation metrics are given in Appendix F.4.

The $n$-d complex hyperbolic embeddings have around double parameters of the $n$-d real embeddings since the $n$-d complex hyperbolic vectors have $n$-d real part and $n$-d imaginary part. For a fair comparison, in each experimental setting, we compare our $n$-d complex hyperbolic embeddings of UnitBall against the $2n$-d embeddings of the baselines. The results will also demonstrate that the $n$-d complex hyperbolic space is not simply the $2n$-d hyperbolic space, they have different capacities.

### 5.2 GRAPH RECONSTRUCTION

### 5.2.1 RESULTS ON COMPRESSED GRAPHS

To illustrate the benefits of UnitBall on varying hierarchical structures, we evaluate on the synthetic compressed graphs. The compressed graphs have local tree structures while being much more complicated than trees. Each compressed graph-$(m, k)$ consists of $m$ nodes and is aggregated from $k$ random trees on the $m$ nodes. The bigger $k$ corresponds to the denser and noisier graph.

Figure 1 depicts that the graph reconstruction results drop down with the increase of $m$ and $k$, which represents the increase of graph scale and denseness respectively. Remarkably, UnitBall outperforms all baselines on the challenging data, showing that UnitBall handles the noisy locally tree-like structures better. The construction-based method TreeRep learns a tree structure from the data as an intermediate step and then embeds the learned trees into the hyperbolic space using Sarkar's construction (Sarkar, 2011). TreeRep has comparable results with other methods when $(m, k) = (500, 1)$ since when $k = 1$, the graph is exactly a tree, i.e., $\delta = 0$. However, when $k > 1$ and $\delta > 0$, the data metrics deviate from tree metrics, in which case it does not help much to learn a tree structure from the data as an intermediate step.

### 5.2.2 RESULTS ON MULTITREE STRUCTURE

In this section, we compare the performances of the hyperbolic models and UnitBall on Xiphophorus. Xiphophorus is a multitree dataset that is formed of 160 mrbayes consensus trees on 26 Xiphophorus fishes. Its cloud tree plot is in Figure 2. The multitree structure is widely used to represent multiple overlapping taxonomies over the same ground set (please refer to Appendix E for details). We reconstruct the edges containing the leaf nodes on Xiphophorus since the leaf links have the practical taxonomic meaning. The results are reported in the last column of Table 4. The results show that the complex hyperbolic geometry has a stronger ability to represent the multitree structure.

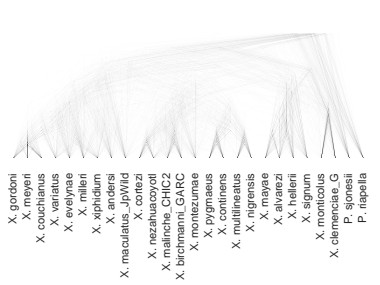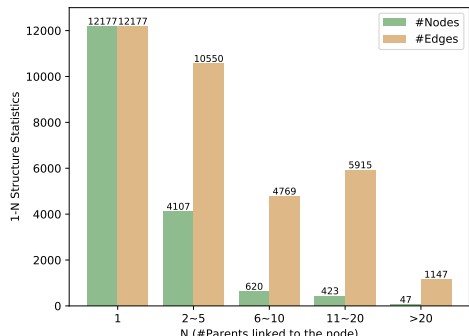

Figure 2: **Left**: The cloud tree plot of Xiphophorus. **Right**: The 1-$N$ structure statistics in YAGO3-wikiObjects *original* taxonomy. The horizontal axis represents the number of parents linked to a node. The vertical axis counts the number of nodes and edges in the 1-$N$ structures. For example, there are $4,107$ nodes that link to $N$ parents for $2 \leq N \leq 5$, and these links count to $10,550$ edges.

Table 2: Evaluation of taxonomy link prediction in 32-d embedding spaces (16-d complex hyperbolic space for UnitBall). The best results are shown in boldface. The second best results are underlined.

| | ICD10 | | | YAGO3-wikiObjects | | | WordNet-noun | | |
|---|---|---|---|---|---|---|---|---|---|
| | MAP | MRR | Hits@3 | MAP | MRR | Hits@3 | MAP | MRR | Hits@3 |
| Euclidean | 3.75 | 3.72 | 2.39 | 4.85 | 4.45 | 2.78 | 5.59 | 5.36 | 3.16 |
| TreeRep | 4.96 | 7.92 | 8.49 | 20.19 | 21.85 | 27.19 | 9.30 | 9.98 | 11.90 |
| Poincaré | 35.24 | 34.45 | 52.71 | 30.06 | 28.47 | 41.61 | 25.46 | 23.99 | 27.80 |
| Hyperboloid | 34.80 | 34.01 | 52.88 | 30.80 | 29.21 | 43.17 | 25.65 | 24.15 | 27.50 |
| UnitBall | **47.88** | **46.96** | **70.28** | **33.33** | **31.85** | **47.41** | **27.29** | **25.93** | **32.95** |

## 5.3 LINK PREDICTION

### 5.3.1 RESULTS ON THE REAL-WORLD TAXONOMIES

In this section, we evaluate the performances on the link prediction task for real-world taxonomies. Table 2 presents the results in 32-d embedding spaces for baselines and 16-d complex hyperbolic space for UnitBall. Predicting missing links requires generalization capacity, and UnitBall still has the best performances on all three datasets. Besides, we see that Euclidean shows shortages on these hierarchically-structured data, which is consistent with the results in previous works (Nickel and Kiela, 2017; 2018). Similar to the results on the graph reconstruction task, Poincaré and Hyperboloid have very close performances, while Hyperboloid has slightly better results. They have significant improvements over Euclidean, but they still fall behind UnitBall, which demonstrates our claims that the non-constant negative curvature of the complex hyperbolic space addresses the varying hierarchical structures on real-world taxonomies.

We notice that TreeRep does not perform well on the link prediction task. As mentioned in Section 2, the combinatorial construction-based embedding methods target minimizing the reconstruction distortion of data. However, minimizing the reconstruction distortion may overfit the training set, thus resulting in the unpromising generalization performance for unobserved edges. Hence, they are more suitable to learn the representation of graph data without missing links. The evaluation of TreeRep on the graph reconstruction task can be referred to Section 5.2.1, Appendix F.5 and F.6.

### 5.3.2 EXPLORING THE EMBEDDING DIMENSIONS

In this section, we explore the performances in different embedding dimensions. The results on YAGO3-wikiObjects are presented in Table 3. Results on other datasets are in Appendix F.7. We find that with the increase of the embedding dimension, Euclidean can have big improvements, but its performances in 128-d still cannot surpass other methods in 8-d. TreeRep also achieves better results with the increase of dimension, but overall its performances on the link prediction task are not very promising. By comparison, Poincaré, Hyperboloid, and UnitBall achieve great results steadily. 8-d is already enough for Poincaré and Hyperboloid to handle the link prediction task while

Table 3: Evaluation of taxonomy link prediction in different embedding dimensions (the embedding dimension for UnitBall is half of other models). The best results are shown in boldface. The second best results are underlined.

| | YAGO3-wikiObjects | | | | | | | | |
| | 8-dimensional | | | 32-dimensional | | | 128-dimensional | | |
| | MAP | MRR | Hits@3 | MAP | MRR | Hits@3 | MAP | MRR | Hits@3 |
| Euclidean | 1.02 | 0.92 | 0.57 | 4.85 | 4.45 | 2.78 | 16.67 | 15.76 | 15.97 |
| TreeRep | 16.91 | 17.48 | 27.53 | 20.19 | 21.85 | 27.19 | 21.18 | 23.44 | 32.84 |
| Poincaré | 29.70 | 28.13 | 41.64 | 30.06 | 28.47 | 41.61 | 29.93 | 28.35 | 41.53 |
| Hyperboloid | 30.87 | 29.28 | 43.50 | 30.80 | 29.21 | 43.17 | 30.68 | 29.07 | 42.86 |
| UnitBall | **31.40** | **29.98** | **44.25** | **33.33** | **31.85** | **47.41** | **32.76** | **31.28** | **46.25** |

Table 4: Results of Hits@10 on **predicting** $1$-$N$ **edges in YAGO3-wikiObjects** and results of MAP on **reconstructing the leaf node links in Xiphophorus**. The embedding dimension is 16 for UnitBall and 32 for other models. The best results are shown in boldface.

| | YAGO3-wikiObjects (Hits@10) | | | | | | Xiphophorus (MAP) |
| $N$ for 1-$N$ edges | 1 | $2 \sim 5$ | $6 \sim 10$ | $11 \sim 20$ | $> 20$ | $> 1$ | Leaf Links Reconstruction |
| Poincaré | **93.10** | 65.48 | 60.11 | 49.09 | 43.97 | 63.29 | 89.75 |
| Hyperboloid | 92.80 | 65.51 | 63.49 | 51.93 | 45.39 | 63.95 | 89.80 |
| UnitBall | 92.42 | **76.35** | **65.91** | **66.19** | **70.21** | **74.03** | **91.95** |

UnitBall has small improvements from $4$-d to $16$-d, then converges to the stable performance. The results demonstrate that the Euclidean embeddings need to increase the dimension to better model the hierarchies, while the complex hyperbolic space and the hyperbolic space have strong generalization competence for hierarchical structures.

### 5.3.3 Exploring 1-$N$ Structure

A noteworthy difference between the real-world taxonomies and the tree structures is that the taxonomies contain many $1$-$N$ (1 child links to multiple parents) cases while in a tree each node except the root is linked to only 1 parent node. To investigate the advantages of complex hyperbolic geometry on the $1$-$N$ structures, we evaluate the performances of UnitBall, Poincaré, and Hyperboloid on predicting the $1$-$N$ edges. We evaluate on YAGO3-wikiObjects since it contains abundant $1$-$N$ structures. The statistics of the $1$-$N$ structures in YAGO3-wikiObjects *original* taxonomy (*original* means without computing the transitive closure) are given in Figure 2.

In this experiment, we train the embeddings on the full transitive closure of YAGO3-wikiObjects and then predict the $1$-$N$ edges. The Hits@10 scores are reported on Table 4. For results of other metrics, please refer to Appendix F.8. We can see that UnitBall has a very small compromise for the 1-1 edges, i.e., the edge pattern of tree structures. Furthermore, UnitBall outperforms the hyperbolic models largely on $1$-$N$ structures for $N > 1$. Even for nodes that link to more than 20 parents, UnitBall can have accurate top 10 predictions for these edges. The results demonstrate that the complex hyperbolic embeddings maintain the advantages in the edge pattern of tree structures as well as handling more complicated hierarchical structures compared with the real hyperbolic embeddings.

## 6 Conclusion and Future Work

In this paper, we present a novel approach for learning the embeddings of hierarchical structures in the unit ball model of the complex hyperbolic space. We characterize the geometrical properties of the complex hyperbolic space and formulate the embedding algorithm in the unit ball model. We exemplify the superiority of our approach over the graph reconstruction task and the link prediction task on both synthetic and real-world data, which cover the various hierarchical structures and two specific structures, namely multitree structure and $1$-$N$ structure. The empirical results show that our approach outperforms the hyperbolic embedding methods in terms of representation capacity and generalization performance. Motivated by our theoretical grounding and empirical success, we believe the complex hyperbolic embeddings will have promising improvements on the knowledge graph embeddings, neural networks, and other related applications.

## REPRODUCIBILITY STATEMENT

- Please see Section 3 and 4 for the full set of assumptions of all theoretical results. Please see Appendix B and C for complete proofs of all theoretical results.
- Please see Section 5.1 and Appendix F.2, F.3 for the experimental details (e.g., data splits, hyperparameters, hardware, etc).
- The data used in the paper are publicly available. We cite the corresponding references and give the public data links in Appendix F.1.
- The code is proprietary for this moment. The code will be released after the the publication of the paper.

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

## A   MORE PRELIMINARIES

### A.1   DEFINITION OF CURVATURE

The *curvature* describes the curve of Riemannian manifolds and controls the rate of geodesic deviation. In this paper, *curvature* refers to the *sectional curvature*.

**Definition 1** (Curvature). *Given a Riemannian manifold and two linearly independent tangent vectors at the same point,* $\mathbf{u}$ *and* $\mathbf{v}$, *the (sectional) curvature is defined as*

$$K(\mathbf{u}, \mathbf{v}) = \frac{\langle R(\mathbf{u}, \mathbf{v})\mathbf{v}, \mathbf{u} \rangle}{\langle \mathbf{u}, \mathbf{u} \rangle \langle \mathbf{v}, \mathbf{v} \rangle - \langle \mathbf{u}, \mathbf{v} \rangle^2},$$

*where* $R$ *is the Riemann curvature tensor, defined by the convention* $R(\mathbf{u}, \mathbf{v})\mathbf{w} = \nabla_{\mathbf{u}}\nabla_{\mathbf{v}}\mathbf{w} - \nabla_{\mathbf{v}}\nabla_{\mathbf{u}}\mathbf{w} - \nabla_{[\mathbf{u},\mathbf{v}]}\mathbf{w}$. *Here* $\nabla$ *indicates the Levi-Civita connection, whose definitions are given below.*

Before defining Levi-Civita connection, we need to first define the affine connection.

**Definition 2** (Affine connection). *Let* $M$ *be a smooth manifold and let* $\Gamma(\mathcal{T}M)$ *be the space of vector fields on* $M$, *i.e., the space of smooth sections of the tangent bundle* $\mathcal{T}M$. *Then an **affine connection** on M is a bilinear map*

$$\Gamma(\mathcal{T}M) \times \Gamma(\mathcal{T}M) \to \Gamma(\mathcal{T}M)$$
$$(X, Y) \mapsto \nabla_X Y,$$

*such that for all* $f$ *in the set of smooth functions on* $M$, *written* $C^\infty(M, \mathbb{R})$, *and all vector fields* $X, Y$ *on* $M$:

1. $\nabla_{fX} Y = f\nabla_X Y$, *i.e.,* $\nabla$ *is* $C^\infty(M, \mathbb{R})$-*linear in the first variable;*

2. $\nabla_X(fY) = \partial_X fY + f\nabla_X Y$, *where* $\partial_X$ *denotes the directional derivative, i.e.,* $\nabla$ *satisfies Leibniz rule in the second variable.*

Next, we define the Levi-Civita connection.

**Definition 3** (Levi-Civita connection). *An affine connection* $\nabla$ *is called a **Levi-Civita** connection if*

1. *it preserves the metric, i.e.,* $\nabla g = 0$.

2. *it is torsion-free, i.e., for any vector fields* $X$ *and* $Y$ *we have* $\nabla_X Y - \nabla_Y X = [X, Y]$, *where* $[X, Y]$ *is the Lie bracket of the vector fields* $X$ *and* $Y$.

### A.2   THE DISTANCE FUNCTION ON THE UNIT BALL MODEL

The distance function in Eq. (9) maintains the tree-like metric properties. When the points are very close to the origin, it approximates to the Euclidean distance. Additionally, when a point is closer to the origin, it has relatively smaller distances to the other points. Correspondingly, the points near the boundary have very large distances from each other. Therefore, in ideal conditions, the root node of a tree is embedded in the origin while the deeper nodes are embedded farther away from the origin. Recall that the distance function in the real hyperbolic space (Nickel and Kiela, 2017) has similar properties since the real hyperbolic space, as the totally geodesic subspace of the complex hyperbolic space, inherits the tree-like metrics. More details about the Bergman metric and distance function can be referred to Chapter 3.1 in (Goldman, 1999).

## B   PROOF OF THEOREM 1

In Section 3.2 in the paper, we presented Theorem 1 about the curvature of the complex hyperbolic space $\mathbb{H}^n_{\mathbb{C}}$ (Goldman, 1999). The sketch explanation is that all unit tangent vectors are equivalent, but not all directions are spanned by two unit tangent vectors. Before proving Theorem 1, we need to introduce the definition of *Kähler structure* (Mok, 1989).

**Definition 4** (Kähler structure). *A **Kähler structure** can be defined in any of the following equivalent ways:*

1. *A complex structure with a closed, positive $(1,1)$-form.*

2. *A Riemannian structure with a complex structure such that the corresponding exterior 2-form is closed.*

3. *A symplectic structure with a compatible integrable almost complex structure which is positive.*

Recall that in Section 3.2, we defined the complex hyperbolic space $\mathbb{H}^n_{\mathbb{C}}$ using the projectivization of the negative zone with a Hermitian form $\langle\langle \mathbf{z}, \mathbf{w} \rangle\rangle$. Denote $\omega$ as the imaginary part of the Hermitian form $\langle\langle , \rangle\rangle$, i.e., $\omega(\mathbf{z}, \mathbf{w}) = \frac{1}{2i}(\langle\langle \mathbf{z}, \mathbf{w} \rangle\rangle - \langle\langle \mathbf{w}, \mathbf{z} \rangle\rangle)$, then according to (Goldman, 1999), the metric $\omega$ is *positive* and *closed*, and necesssarily has type $(1,1)$. Then by the first definition in Definition 4, $\mathbb{H}^n_{\mathbb{C}}$ is a Kähler structure.

Let $M$ be a Kähler manifold and $\mathbf{z} \in M$. Denote $\mathcal{T}_{\mathbf{z}}M$ as the tangent space of $M$ at $\mathbf{z}$ and $J : \mathcal{T}M \to \mathcal{T}M$ is an endomorphism. As proved in (Kobayashi and Nomizu, 1963), the curvature of real 2-planes in the tangent space $\mathcal{T}_{\mathbf{z}}M$ has the following properties:

**Theorem 2.** *Let $M$ be a connected Kähler manifold of complex dimension $n \geq 2$. If the holomorphic sectional curvature $K(p)$, where $p$ is a plane in $\mathcal{T}_{\mathbf{z}}M$ invariant by $J$, depends only on $\mathbf{z}$, then $M$ is a space of constant holomorphic sectional curvature.*

Next, we give a proposition in (Kobayashi and Nomizu, 1963), which is about the curvature of a plane.

**Proposition 1.** *If $\mathbf{u}, \mathbf{v}$ is an orthonormal basis for a plane $p$ and if we set the curvature of $p$ as $K(p) = R(\mathbf{u}, \mathbf{v})$, where $R(\mathbf{u}, \mathbf{v})$ is the Riemann curvature tensor, then*

$$K(p) = \frac{1}{4}(1 + 3\cos^2 \alpha(p)),$$

*where $\alpha(p)$ is the angle between $p$ and $J(p)$.*

Finally, we prove Theorem 1 as follows.

*Proof.* Let $M$ be a Kähler manifold and $\mathbf{z} \in M$. From Theorem 2, the corresponding sectional curvature function of real 2-planes in $\mathcal{T}_{\mathbf{z}}M$ is completely determined by the sectional curvature function restricted to complex lines in $\mathcal{T}_{\mathbf{z}}M$. If the sectional curvature of every complex line in $\mathcal{T}M$ equals $\kappa$, then $M$ is said to have constant holomorphic sectional curvature $\kappa$.

Then from Proposition 1, we can know that in this case, the sectional curvature of a 2-dimensional subspace $S \subset \mathcal{T}M$ is

$$K(S) = \kappa \frac{1 + 3\cos^2 \alpha(S)}{4}, \tag{17}$$

where $\alpha(S)$ is the angle of holomorphy, defined as the smallest angle between two nonzero vectors from two linear subspaces of the underlying real vector space of $M$.

In particular, the complex hyperbolic space $\mathbb{H}^n_{\mathbb{C}}$ is a Kähler structure with $\kappa = -1$. Since $0 \leq \cos^2 \alpha(S) \leq 1$, then from Eq. (17), we can have $-1 \leq K(S) \leq -1/4$ for any 2-dimensional subspace $S \subset \mathcal{T}M$ of $\mathbb{H}^n_{\mathbb{C}}$, i.e., the (sectional) curvature is not constant in $\mathbb{H}^n_{\mathbb{C}}$, but pinched between $-1$ and $-1/4$. Thus we proved the non-constant curvature of $\mathbb{H}^n_{\mathbb{C}}$.

Specifically, we discuss the complex projective lines and totally real planes in the unit ball model of the complex hyperbolic space:

$$\mathcal{B}^n_{\mathbb{C}} = \{(z_1, \cdots, z_n, 1) | |z_1|^2 + \cdots + |z_n|^2 < 1\}. \tag{18}$$

First let's consider the case of complex projective lines. Consider a complex line $L$ in $\mathbb{C}^n$ that intersects the unit ball model $\mathcal{B}^n_{\mathbb{C}}$. Let $\mathbf{z}$ be any point in $L \cap \mathcal{B}^n_{\mathbb{C}}$. We can apply an element of $PU(n, 1)$ to $L$ so that it becomes the last coordinate axis $\{(\mathbf{0}, z_n) | z_n \in \mathbb{C}\}$, whose intersection with $\mathcal{B}^n_{\mathbb{C}}$ is the disk $|z_n| < 1$. Then the restriction of the Bergman metric to this disc is the Poincaré metric (Beardon, 2012) of constant curvature $-1$.

In order to see this, let $\mathbf{z} = (\mathbf{0}, z_n, 1)$ and $\mathbf{w} = (\mathbf{0}, w_n, 1)$, $\mathbf{z}, \mathbf{w} \in L \cap \mathcal{B}_{\mathbb{C}}^n$, then from Eq. (9) in Section 4.1, the distance between $\mathbf{z}$ and $\mathbf{w}$ is given by Eq. (9). Then we have

$$\cosh^2\left(\frac{d_{\mathcal{B}_{\mathbb{C}}^n}(\mathbf{z}, \mathbf{w})}{2}\right) = \frac{\langle\!\langle \mathbf{z}, \mathbf{w} \rangle\!\rangle \langle\!\langle \mathbf{w}, \mathbf{z} \rangle\!\rangle}{\langle\!\langle \mathbf{z}, \mathbf{z} \rangle\!\rangle \langle\!\langle \mathbf{w}, \mathbf{w} \rangle\!\rangle} = \frac{|z_n \overline{w_n} - 1|^2}{(|z_n|^2 - 1)(|w_n|^2 - 1)}, \tag{19}$$

which is just the Poincaré metric (Beardon, 2012).

Next consider a totally real plane $p$. Any totally real plane $p$ is the image under an element of $PU(n, 1)$ of the subspace comprising those points of $\mathcal{B}_{\mathbb{C}}^n$ with real coordinates, that is actually an embedded copy of the real hyperbolic space $\mathbb{H}_{\mathbb{R}}^n = \{(x_1, \ldots, x_n) | x_1, \ldots, x_n \in \mathbb{R}\}$. This subspace intersects $\mathcal{B}_{\mathbb{C}}^n$ in the subset consisting of those points with $x_1^2 + \cdots + x_n^2 < 1$. Then the Bergman metric restricted to this real-space unit ball is just the Klein-Beltrami metric (Ratcliffe et al., 1994) on the unit ball in $\mathbb{R}^n$ with constant curvature $-1/4$.

To see this, let $\mathbf{x} = (x_1, \ldots, x_n, 1)$ and $\mathbf{y} = (y_1, \ldots, y_n, 1)$, $\mathbf{x}, \mathbf{y} \in \mathbb{H}_{\mathbb{R}}^n \cap \mathcal{B}_{\mathbb{C}}^n$, then apply the similar process with the above, we have

$$\cosh^2\left(\frac{d_{\mathcal{B}_{\mathbb{C}}^n}(\mathbf{x}, \mathbf{y})}{2}\right) = \frac{\langle\!\langle \mathbf{x}, \mathbf{y} \rangle\!\rangle \langle\!\langle \mathbf{y}, \mathbf{x} \rangle\!\rangle}{\langle\!\langle \mathbf{x}, \mathbf{x} \rangle\!\rangle \langle\!\langle \mathbf{y}, \mathbf{y} \rangle\!\rangle} = \frac{(x_1 y_1 + \cdots + x_n y_n - 1)^2}{(x_1^2 + \cdots + x_n^2 - 1)(y_1^2 + \cdots + y_n^2 - 1)}, \tag{20}$$

which is the Klein-Beltrami metric (Ratcliffe et al., 1994) on the unit ball in $\mathbb{R}^n$ with constant curvature $-1/4$.

Therefore, we proved that the curvature of $\mathbb{H}_{\mathbb{C}}^n$ is $-1$ in the directions of complex projective lines while $-1/4$ in the directions of totally real planes. $\square$

**Remark.** *Curvature in the complex hyperbolic space is a very complicated topic in geometric group theory and differential geometry. The complex projective lines and the totally real planes are two kinds of special subspaces, whose curvatures are presented above. For the subspace that lives in between the two special cases, its curvature can be computed accordingly. Since digging into all the curvature details of the complex hyperbolic geometry is not the essential part of our work, we have omitted the related content in our paper. We refer the interested readers to (Fisher) for an interesting example in the complex hyperbolic space (its curvature differs with our work with a constant multiplier 4). In Section 5 and Figure 5 of (Fisher), the author explored the curvature of a triangle in complex hyperbolic geometry with numerical computation.*

## C   DERIVATION OF DISTANCE GRADIENT IN THE UNIT BALL MODEL

The distance function in the unit ball model is given by Eq. (9). We need to compute the distance gradient $\nabla_E d_{\mathcal{B}_{\mathbb{C}}^n}(\mathbf{z}, \mathbf{w}) = \frac{\partial d_{\mathcal{B}_{\mathbb{C}}^n}(\mathbf{z}, \mathbf{w})}{\partial \mathbf{x}} + i \frac{\partial d_{\mathcal{B}_{\mathbb{C}}^n}(\mathbf{z}, \mathbf{w})}{\partial \mathbf{y}}$ during the Riemannian optimization. The full derivation is as follows.

First, we need to introduce Wirtinger derivatives (Wirtinger, 1927), which constructs a differential calculus for differential functions on complex domains.

**Definition 5** (Wirtinger derivatives). *The partial derivatives of a (complex) function $f(z)$ of a complex variable $z = x + iy \in \mathbb{C}, x, y \in \mathbb{R}$, with respect to $z$ and $\bar{z} = x - iy$, respectively, are defined as:*

$$\frac{\partial f(z, \bar{z})}{\partial z} = \frac{1}{2}\left(\frac{\partial}{\partial x} - i\frac{\partial}{\partial y}\right)f(z, \bar{z}), \qquad \frac{\partial f(z, \bar{z})}{\partial \bar{z}} = \frac{1}{2}\left(\frac{\partial}{\partial x} + i\frac{\partial}{\partial y}\right)f(z, \bar{z}).$$

The Wirtinger derivatives can be rewritten as:

$$\frac{\partial f(z, \bar{z})}{\partial x} = \left(\frac{\partial}{\partial z} + \frac{\partial}{\partial \bar{z}}\right)f(z, \bar{z}), \tag{21}$$

$$\frac{\partial f(z, \bar{z})}{\partial y} = i\left(\frac{\partial}{\partial z} - \frac{\partial}{\partial \bar{z}}\right)f(z, \bar{z}), \tag{22}$$

Let $p = \cosh(d_{\mathcal{B}_{\mathbb{C}}^n}(\mathbf{z}, \mathbf{w})) = 2 \frac{\langle\!\langle \mathbf{z}, \mathbf{w} \rangle\!\rangle \langle\!\langle \mathbf{w}, \mathbf{z} \rangle\!\rangle}{\langle\!\langle \mathbf{z}, \mathbf{z} \rangle\!\rangle \langle\!\langle \mathbf{w}, \mathbf{w} \rangle\!\rangle} - 1$, then $d_{\mathcal{B}_{\mathbb{C}}^n}(\mathbf{z}, \mathbf{w}) = arcosh(p) = \ln(p + \sqrt{p^2 - 1})$.
Let $\mathbf{z} = (z_1, \ldots, z_n, 1) \in \mathcal{B}_{\mathbb{C}}^n$, then

$$
\begin{aligned}
\frac{\partial d_{\mathcal{B}_{\mathbb{C}}^n}(\mathbf{z}, \mathbf{w})}{\partial z_j} &= \frac{\partial d_{\mathcal{B}_{\mathbb{C}}^n}(\mathbf{z}, \mathbf{w})}{\partial p} \cdot \frac{\partial p}{\partial z_j} = \frac{1}{\sqrt{p^2 - 1}} \cdot \frac{\partial p}{\partial z_j} \\
&= \frac{2}{\sqrt{p^2 - 1}} \cdot \frac{\partial \frac{(z_1 \overline{w_1} + \cdots + z_n \overline{w_n} - 1) \cdot \langle\!\langle \mathbf{w}, \mathbf{z} \rangle\!\rangle}{(z_1 \overline{z_1} + \cdots + z_n \overline{z_n} - 1) \cdot \langle\!\langle \mathbf{w}, \mathbf{w} \rangle\!\rangle}}{\partial z_j} \\
&= \frac{2}{\sqrt{p^2 - 1}} \cdot \left( \frac{\overline{w_j} \langle\!\langle \mathbf{w}, \mathbf{z} \rangle\!\rangle}{\langle\!\langle \mathbf{z}, \mathbf{z} \rangle\!\rangle \cdot \langle\!\langle \mathbf{w}, \mathbf{w} \rangle\!\rangle} - \frac{\overline{z_j} \langle\!\langle \mathbf{z}, \mathbf{w} \rangle\!\rangle \cdot \langle\!\langle \mathbf{w}, \mathbf{z} \rangle\!\rangle}{\langle\!\langle \mathbf{z}, \mathbf{z} \rangle\!\rangle^2 \cdot \langle\!\langle \mathbf{w}, \mathbf{w} \rangle\!\rangle} \right),
\end{aligned} \tag{23}
$$

for $1 \leq j \leq n$. Similarly, we can have

$$
\frac{\partial d_{\mathcal{B}_{\mathbb{C}}^n}(\mathbf{z}, \mathbf{w})}{\partial \overline{z_j}} = \frac{2}{\sqrt{p^2 - 1}} \cdot \left( \frac{w_j \langle\!\langle \mathbf{z}, \mathbf{w} \rangle\!\rangle}{\langle\!\langle \mathbf{z}, \mathbf{z} \rangle\!\rangle \cdot \langle\!\langle \mathbf{w}, \mathbf{w} \rangle\!\rangle} - \frac{z_j \langle\!\langle \mathbf{z}, \mathbf{w} \rangle\!\rangle \cdot \langle\!\langle \mathbf{w}, \mathbf{z} \rangle\!\rangle}{\langle\!\langle \mathbf{z}, \mathbf{z} \rangle\!\rangle^2 \cdot \langle\!\langle \mathbf{w}, \mathbf{w} \rangle\!\rangle} \right). \tag{24}
$$

Then by Eqs. (21), (23), and (24), we obtain

$$
\frac{\partial d_{\mathcal{B}_{\mathbb{C}}^n}}{\partial x_j} = \frac{\partial d_{\mathcal{B}_{\mathbb{C}}^n}(\mathbf{z}, \mathbf{w})}{\partial z_j} + \frac{\partial d_{\mathcal{B}_{\mathbb{C}}^n}(\mathbf{z}, \mathbf{w})}{\partial \overline{z_j}} = \frac{4}{\sqrt{p^2 - 1}} \left( \frac{Re(\langle\!\langle \mathbf{z}, \mathbf{w} \rangle\!\rangle w_j)}{\langle\!\langle \mathbf{z}, \mathbf{z} \rangle\!\rangle \langle\!\langle \mathbf{w}, \mathbf{w} \rangle\!\rangle} - \frac{\langle\!\langle \mathbf{z}, \mathbf{w} \rangle\!\rangle \langle\!\langle \mathbf{w}, \mathbf{z} \rangle\!\rangle x_j}{\langle\!\langle \mathbf{z}, \mathbf{z} \rangle\!\rangle^2 \langle\!\langle \mathbf{w}, \mathbf{w} \rangle\!\rangle} \right). \tag{25}
$$

Similarly, by Eqs. (22), (23), and (24), we can get

$$
\frac{\partial d_{\mathcal{B}_{\mathbb{C}}^n}}{\partial y_j} = i \left( \frac{\partial d_{\mathcal{B}_{\mathbb{C}}^n}(\mathbf{z}, \mathbf{w})}{\partial z_j} - \frac{\partial d_{\mathcal{B}_{\mathbb{C}}^n}(\mathbf{z}, \mathbf{w})}{\partial \overline{z_j}} \right) = \frac{4}{\sqrt{p^2 - 1}} \left( \frac{Im(\langle\!\langle \mathbf{z}, \mathbf{w} \rangle\!\rangle w_j)}{\langle\!\langle \mathbf{z}, \mathbf{z} \rangle\!\rangle \langle\!\langle \mathbf{w}, \mathbf{w} \rangle\!\rangle} - \frac{\langle\!\langle \mathbf{z}, \mathbf{w} \rangle\!\rangle \langle\!\langle \mathbf{w}, \mathbf{z} \rangle\!\rangle y_j}{\langle\!\langle \mathbf{z}, \mathbf{z} \rangle\!\rangle^2 \langle\!\langle \mathbf{w}, \mathbf{w} \rangle\!\rangle} \right), \tag{26}
$$

where $Re(\cdot)$ and $Im(\cdot)$ denote the real and the imaginary part respectively. Then we can have

$$
\begin{aligned}
\frac{\partial d_{\mathcal{B}_{\mathbb{C}}^n}}{\partial \mathbf{x}} &= \frac{4}{\sqrt{p^2 - 1}} \left( \frac{Re(\langle\!\langle \mathbf{z}, \mathbf{w} \rangle\!\rangle \mathbf{w})}{\langle\!\langle \mathbf{z}, \mathbf{z} \rangle\!\rangle \langle\!\langle \mathbf{w}, \mathbf{w} \rangle\!\rangle} - \frac{\langle\!\langle \mathbf{z}, \mathbf{w} \rangle\!\rangle \langle\!\langle \mathbf{w}, \mathbf{z} \rangle\!\rangle \mathbf{x}}{\langle\!\langle \mathbf{z}, \mathbf{z} \rangle\!\rangle^2 \langle\!\langle \mathbf{w}, \mathbf{w} \rangle\!\rangle} \right), \\
\frac{\partial d_{\mathcal{B}_{\mathbb{C}}^n}}{\partial \mathbf{y}} &= \frac{4}{\sqrt{p^2 - 1}} \left( \frac{Im(\langle\!\langle \mathbf{z}, \mathbf{w} \rangle\!\rangle \mathbf{w})}{\langle\!\langle \mathbf{z}, \mathbf{z} \rangle\!\rangle \langle\!\langle \mathbf{w}, \mathbf{w} \rangle\!\rangle} - \frac{\langle\!\langle \mathbf{z}, \mathbf{w} \rangle\!\rangle \langle\!\langle \mathbf{w}, \mathbf{z} \rangle\!\rangle \mathbf{y}}{\langle\!\langle \mathbf{z}, \mathbf{z} \rangle\!\rangle^2 \langle\!\langle \mathbf{w}, \mathbf{w} \rangle\!\rangle} \right),
\end{aligned}
$$

which are Eqs. (13) and (14) in Section 4.3.

## D    DEFINITION OF $\delta$-HYPERBOLICITY

In this section, we give the definition of $\delta$-hyperbolicity (Gromov, 1987), which measures the tree-likeness of graphs. The lower $\delta$ corresponds to the more tree-like graph. Trees have $0$ $\delta$-hyperbolicity.

**Definition 6** ($\delta$-hyperbolicity). *Let $a, b, c, d$ be vertices of the graph $G$. Let $S_1$, $S_2$ and $S_3$ be*

$$
S_1 = dist(a, b) + dist(d, c), S_2 = dist(a, c) + dist(b, d), S_3 = dist(a, d) + dist(b, c).
$$

*Suppose $M_1$ and $M_2$ are the two largest values among $S_1$, $S_2$, $S_3$ and $M_1 \geq M_2$. Define $hyp(a, b, c, d) = M_1 - M_2$. Then the $\delta$-**hyperbolicity** of $G$ is defined as*

$$
\delta(G) = \frac{1}{2} \max_{a, b, c, d \in V(G)} hyp(a, b, c, d).
$$

*That is, $\delta(G)$ is the maximum of $hyp$ over all possible $4$-tuples $(a, b, c, d)$ divided by 2.*

## E    MULTITREE STRUCTURE

In combinatorics and order-theoretic mathematics, a multitree structure is a directed acyclic graph (DAG) in which the set of vertices reachable from any vertex induces a tree, or a partially ordered set (poset) that does not have four items $a$, $b$, $c$, and $d$ forming a diamond suborder with $a \leq b \leq d$ and

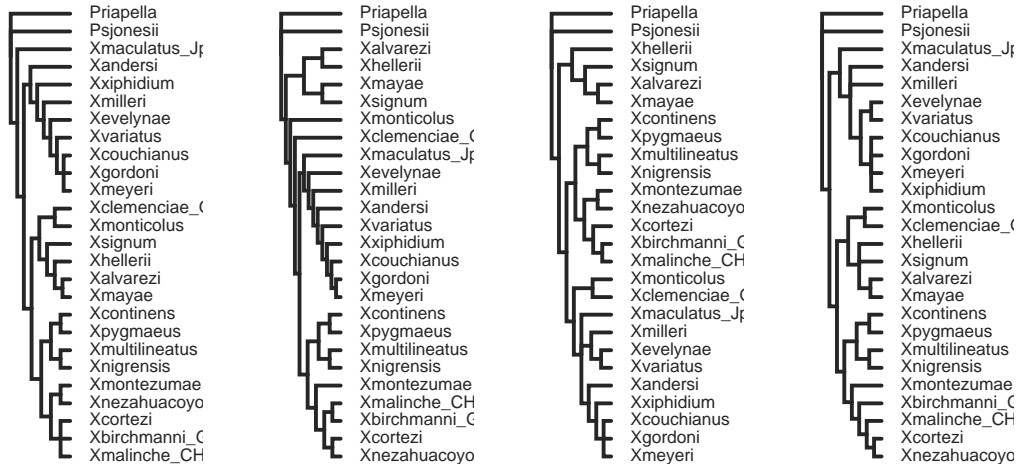

Figure 3: Some subtrees of Xiphophorus. The multitree dataset Xiphophorus contains 160 trees on 26 Xiphophorus fishes as leaf nodes.

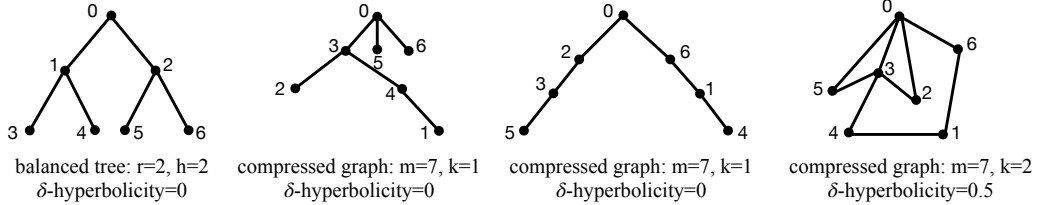

| balanced tree: r=2, h=2 | compressed graph: m=7, k=1 | compressed graph: m=7, k=1 | compressed graph: m=7, k=2 |
| $\delta$-hyperbolicity=0 | $\delta$-hyperbolicity=0 | $\delta$-hyperbolicity=0 | $\delta$-hyperbolicity=0.5 |

Figure 4: Simple examples of the synthetic data. The numbers $\{0, 1, \ldots, 6\}$ represent the nodes. The compressed graph-($m = 7$, $k = 2$) on the right are aggregated from the middle two compressed graphs-($m = 7$, $k = 1$).

$a \leq c \leq d$ but with $b$ and $c$ incomparable to each other (also called a diamond-free poset (Griggs et al., 2012)).[4]

Obviously, the multitree structure is not a tree since one child node can have multiple parents in multitree. Note that the multitree structure is also different with 1-$N$ structure since the multitree has more strict conditions, that is, the multitree is a diamond-free poset. By comparison, the 1-$N$ structure is more general in taxonomies. For example, the subgraph of YAGO3-wikiObjects {(*Nei Gaiman*, *is-a*, *British screenwriters*), (*Neil Gaiman*, *is-a*, *British fantasy writers*), (*British screenwriters*, *is-a*, *British writers*), (*British fantasy writers*, *is-a*, *British writers*)} is a 1-$N$ structure, but it is not a multitree structure, because *British screenwriters* and *British fantasy writers* are incomparable to each other, i.e., there is no partial order between them.

Recall that our synthetic compressed graph is also aggregated from multiple trees, but it is not the multitree either. A compressed graph is aggregated from multiple random trees on the same set of nodes while the trees in a multitree structure share the same leaf nodes. Multitrees are widely used to represent multiple overlapping taxonomies over the same ground set. The dataset Xiphophorus is a multitree structure containing 160 trees on 26 leaf nodes. Some examples of its subtrees are in Figure 3 and its cloud tree plot is in Figure 2.

## F    MORE EXPERIMENTS

### F.1    DATA

In Section 5.1.1 in the paper, we introduced how we generate the synthetic data:

---

[4]Here $\leq$ denotes the partial order defined in the graph, e.g., the hypernymy relation.

Table 5: Hyperparameters of all methods.

| Model | Synthetic & Xiphophorus | ICD10 | YAGO3-wikiObjects | WordNet-noun |
|---|---|---|---|---|
| TreeRep | iterations: 20; optimization: *no opt*; pre-allocation fraction: 2.0; nthreads: 16; terminated edge weight: 0; trials/dataset: 3 | iterations: 32; optimization: *no opt*; pre-allocation fraction: 1.3; nthreads: 16; terminated edge weight: 0; trials/dataset: 3 | iterations: 32; optimization: *no opt*; pre-allocation fraction: 1.3; nthreads: 16; terminated edge weight: 0; trials/dataset: 3 | iterations: 1; optimization: *no opt*; pre-allocation fraction: 1.3; nthreads: 16; terminated edge weight: 0; trials/dataset: 3 |
| Euclidean | - | manifold: *euclidean*; learning rate: 1; epochs: 1500; dampening: 0.75; burnin: 20; burnin multiplier: 0.01; negative sample: 50; negative multiplier: 0.1; max norm: 50000 | manifold: *euclidean*; learning rate: 1; epochs: 1200; dampening: 0.75; burnin: 20; burnin multiplier: 0.01; negative sample: 50; negative multiplier: 0.1; max norm: 50000 | manifold: *euclidean*; learning rate: 1; epochs: 1500; dampening: 0.75; burnin: 20; burnin multiplier: 0.01; negative sample: 50; negative multiplier: 0.1; max norm: 50000 |
| Poincaré | manifold: *poincare*; learning rate: 0.3; epochs: 1500; dampening: 0.75; burnin: 20; burnin multiplier: 0.01; negative sample: 50; negative multiplier: 0.1; max norm: $1 - e^{-5}$ | manifold: *poincare*; learning rate: 1; epochs: 1500; dampening: 1.0; burnin: 20; burnin multiplier: 0.01; negative sample: 50; negative multiplier: 0.1; max norm: $1 - e^{-5}$ | manifold: *poincare*; learning rate: 1; epochs: 1200; dampening: 1.0; burnin: 20; burnin multiplier: 0.01; negative sample: 50; negative multiplier: 0.1; max norm: $1 - e^{-5}$ | manifold: *poincare*; learning rate: 1; epochs: 1500; dampening: 1.0; burnin: 20; burnin multiplier: 0.01; negative sample: 50; negative multiplier: 0.1; max norm: $1 - e^{-5}$ |
| Hyperboloid | manifold: *lorentz*; learning rate: 0.3; epochs: 1500; dampening: 0.75; burnin: 20; burnin multiplier: 0.01; negative sample: 50; negative multiplier: 0.1; max norm: *no-maxnorm* | manifold: *lorentz*; learning rate: 0.5; epochs: 1500; dampening: 1.0; burnin: 20; burnin multiplier: 0.01; negative sample: 50; negative multiplier: 0.1; max norm: *no-maxnorm* | manifold: *lorentz*; learning rate: 1; epochs: 1200; dampening: 1.0; burnin: 20; burnin multiplier: 0.01; negative sample: 50; negative multiplier: 0.1; max norm: *no-maxnorm* | manifold: *lorentz*; learning rate: 0.5; epochs: 1500; dampening: 1.0; burnin: 20; burnin multiplier: 0.01; negative sample: 50; negative multiplier: 0.1; max norm: *no-maxnorm* |
| UnitBall | manifold: *unitball*; learning rate: 8; epochs: 1500; dampening: 0.75; burnin: 20; burnin multiplier: 0.01; negative sample: 50; negative multiplier: 0.1; max norm: $1 - e^{-5}$ | manifold: *unitball*; learning rate: 11; epochs: 200; dampening: 1.0; burnin: 20; burnin multiplier: 0.01; negative sample: 50; negative multiplier: 0.1; max norm: $1 - e^{-5}$ | manifold: *unitball*; learning rate: 14; epochs: 1200; dampening: 1.0; burnin: 20; burnin multiplier: 0.01; negative sample: 50; negative multiplier: 0.1; max norm: $1 - e^{-5}$ | manifold: *unitball*; learning rate: 12; epochs: 900; dampening: 1.0; burnin: 20; burnin multiplier: 0.01; negative sample: 50; negative multiplier: 0.1; max norm: $1 - e^{-5}$ |

**Synthetic.** We generate various balanced trees and compressed graphs using the NetworkX package (Hagberg et al., 2008).[5] For **balanced trees**, we generate the balanced tree with degree $r$ and depth $h$. For **compressed graphs**, we generate $k$ random trees on $m$ nodes and then aggregate their edges to form a graph.

We give some examples of the synthetic data in Figure 4. As we can see, the compressed graphs-($m = 7$, $k = 1$) are random trees on 7 nodes, so their $\delta$-hyperbolicities are 0. The compressed graph-($m = 7$, $k = 2$) is no longer a tree after aggregating from two trees. Its local structures are more varying and complicated.

Here we also give the public links of the real-world data used in our experiments: Xiphophorus (Cui et al., 2013),[6] ICD10 (Brämer, 1988),[7] YAGO3-wikiObjects (Mahdisoltani et al., 2015),[8] WordNet-noun (Miller, 1995).[9]

### F.2 HARDWARE

We conduct all the experiments except TreeRep on four NVIDIA GTX 1080Ti GPUs with 8GB memory each. For TreeRep, we need more memory to store the distance matrices, so we use a 96-core NVIDIA T4 GPU server with 503GB memory.

---

[5] https://networkx.org/documentation/stable/reference/generators.html.

[6] https://toytree.readthedocs.io/en/latest/7-multitrees.html.

[7] https://www.who.int/standards/classifications/classification-of-diseases.

[8] https://yago-knowledge.org/.

[9] https://wordnet.princeton.edu/.

### F.3 HYPERPARAMETERS

For the baselines (TreeRep (Sonthalia and Gilbert, 2020),[10] Euclidean, Poincaré (Nickel and Kiela, 2017), and Hyperboloid (Nickel and Kiela, 2018)),[11] we use their public codes to train the embeddings. For all methods, we tune the hyperparameters by grid search. For the graph reconstruction task, we tune the hyperparameters on balanced tree-(15,3) in 20-dimensional embedding spaces (10-dimensional complex hyperbolic space for UnitBall), while for the link prediction task, we tune the hyperparameters on the validation sets in 32-dimensional embedding spaces (16-dimensional complex hyperbolic space for UnitBall). The hyperparameters are given in Table 5.

### F.4 EVALUATION

Our evaluation closely follows the setting of (Nickel and Kiela, 2017; 2018), which infers the hierarchies from distances in the embedding space. Specifically, for each test edge $(z, w)$, we compute the distance between the embeddings $d_{\mathcal{B}^n_\mathbb{C}}(\mathbf{z}, \mathbf{w})$ and rank it among the distances of all unobserved edges for $z$: $\{d_{\mathcal{B}^n_\mathbb{C}}(\mathbf{z}, \mathbf{w}') : (z, w') \notin \text{Training}\}$. We then report the following evaluation metrics of the rankings. Denote $E_{test}$ as the test edge set and $V = \{z | \exists w, (z, w) \in E_{test}\}$ as the test node set. Let $NE_z = \{w_1, w_2, \ldots, w_{|NE_z|}\}$ be the ground truth neighbor set of node $z$.

**Mean average precision (MAP).** The average precision (AP) is a way to summarize the precision-recall curve into a single value representing the average of all precisions and the MAP score is calculated by taking the mean AP over all classes. For a node $z$, from the learned embeddings, we can obtain the nodes closest to its embedding $\mathbf{z}$. Let $R_{z,w_i}$ be the smallest set of such nodes that contains $w_i$ (the $i$-th neighbor of $z$). Then the MAP is defined as:

$$\text{MAP} = \frac{1}{|V|} \sum_{z \in V} \frac{1}{|NE_z|} \sum_{w_i \in NE_z} Precision(R_{z,w_i}).$$

**Mean reciprocal rank (MRR).** The MRR is a statistic measure for evaluating a list of possible responses to a sample of queries, ordered by the probability of correctness. For a node $z$, from the learned embeddings, we can rank its distances with other nodes from the smallest to the largest. Let $rank_{w_i}$ be the rank of $w_i$ (the $i$-th neighbor of $z$). Then the MRR is defined as:

$$\text{MRR} = \frac{1}{|V|} \sum_{z \in V} \frac{1}{|NE_z|} \sum_{w_i \in NE_z} \frac{1}{rank_{w_i}}.$$

**The proportion of correct types that rank no larger than $N$ (Hits@$N$).** Hits@$N$ measures whether the top $N$ predictions contain the ground truth labels. For a node $z$, from the learned embeddings, we can obtain the set of $N$ nodes closest to its embedding $\mathbf{z}$, denoted as $R_z^N$. Then the Hits@$N$ is defined as:

$$\text{Hits@}N = \frac{1}{|V|} \sum_{z \in V} \mathbb{I}(|R_z^N \cap NE_z| \geq 1),$$

where $\mathbb{I}(|R_z^N \cap NE_z| \geq 1)$ is the indicator function.

### F.5 GRAPH RECONSTRUCTION RESULTS ON BALANCED TREES

To compare the representation capacities of UnitBall and the hyperbolic embedding models for the tree structures, we evaluate the graph reconstruction task on the synthetic balanced trees. A balanced tree-$(r, h)$ has degree $r$ and depth $h$, so it has $r^0 + \cdots + r^d$ nodes and $r^0 + \cdots + r^d - 1$ edges. The $\delta$-hyperbolicity of any balanced tree is 0. We embed the balanced trees into 20-d hyperbolic space for the baselines and 10-d complex hyperbolic space for UnitBall.

Figure 5 presents the MAP and Hits@3 scores with varying $r$ and $h$. We see that when the tree is in small scale, e.g., $(r, h) = (15, 3), (10, 2), (10, 3)$, all methods have very good performances,

---

[10]https://github.com/rsonthal/TreeRep.
[11]https://github.com/facebookresearch/poincare-embeddings. The repository provides the implementation for Euclidean, Poincaré, and Hyperboloid.

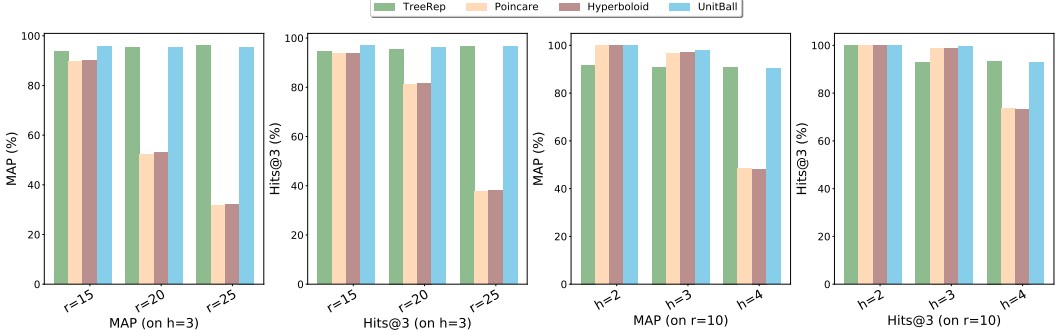

Figure 5: Evaluation of graph reconstruction on synthetic balanced trees in 20-d embedding spaces (10-d complex hyperbolic space for UnitBall). $r$ represents the degree while $h$ represents the depth.

Table 6: Evaluation of graph reconstruction on the real-world taxonomies (the dimension is 32 for TreeRep and 16 for UnitBall). For memory cost, the unit is *GiB*.

| | ICD10 | | | YAGO3-wikiObjects | | | WordNet-noun | | |
|---|---|---|---|---|---|---|---|---|---|
| | MRR | Hits@1 | Memory | MRR | Hits@1 | Memory | MRR | Hits@1 | Memory |
| TreeRep | 26.74 | 91.97 | 30 | 36.71 | 95.39 | 21 | 16.99 | 90.51 | 226 |
| UnitBall | 47.47 | 98.93 | 0.005 | 39.65 | 96.10 | 0.005 | 28.88 | 94.95 | 0.02 |

demonstrating the expected powerful capacities of hyperbolic geometry and complex hyperbolic geometry on tree structures. However, when the breadth or the depth increases, the performances of Poincaré and Hyperboloid drop rapidly, suggesting that the optimization-based embeddings in $\mathbb{H}_{\mathbb{R}}^{20}$ are not effective enough for reconstructing trees of such scales.

In comparison, UnitBall and TreeRep achieve stable performances for larger trees. TreeRep learns a tree structure from the data as an intermediate step and then embeds the learned trees into the hyperbolic space using Sarkar's construction (Sarkar, 2011). When the input data is a tree, TreeRep exactly recovers the original tree structure. Figure 5 shows that UnitBall achieves comparable or even better performances than TreeRep on the balanced trees. The results demonstrate that UnitBall does not compromise on trees. It produces high-quality embeddings for tree structures.

### F.6 COMPARISON WITH TREEREP ON REAL-WORLD TAXONOMY RECONSTRUCTION

In this section, we compare UnitBall with TreeRep on the real-world taxonomy reconstruction task. The results are presented in Table 6. As we analyzed in Section 5.3.1, TreeRep, as a combinatorial construction-based embedding method, is more suitable for the graph reconstruction task. Its performance is much better than that on the link prediction task. In addition, UnitBall still outperforms TreeRep on reconstructing real-world taxonomies.

We also notice the memory issues of the combinatorial construction-based embedding methods. Although TreeRep is very efficient in embedding tree structures since it does not need the gradient-based optimization steps, it costs more memory resources for constructing the tree structures from data. It is basically a computation time vs. memory cost trade-off issue. For a graph with $m$ nodes, TreeRep needs to construct a matrix of size $c \cdot m \times c \cdot m$ to construct the tree structure, where $1 \leq c \leq 2$ is a hyperparameter. We report the memory cost (*GiB*) in Table 6. UnitBall costs much less memory to learn the embeddings.

### F.7 MORE RESULTS ON VARIOUS DIMENSIONS

In Section 5.3.2, we reported the performances in different embedding dimensions on YAGO3-wikiObjects because of the page limits. Here we present the results in 8-d, 32-d, and 128-d embedding spaces (4-d, 16-d and 64-d complex hyperbolic spaces for UnitBall) on ICD10 and WordNet-noun in Table 7. Again, we see that with the increase of the embedding dimension, Euclidean can have big

Table 7: Evaluation of taxonomy link prediction in different embedding dimensions (the embedding dimension for UnitBall is half of other models). The best results are shown in boldface. The second best results are underlined. TreeRep is not applicable to 128-d WordNet-noun due to the large memory cost so we do not include the results.

| | ICD10 | | | | | | | | |
| | 8-dimensional | | | 32-dimensional | | | 128-dimensional | | |
| | MAP | MRR | Hits@3 | MAP | MRR | Hits@3 | MAP | MRR | Hits@3 |
|---|---|---|---|---|---|---|---|---|---|
| Euclidean | 2.57 | 2.57 | 1.32 | 3.75 | 3.72 | 2.39 | 10.83 | 10.48 | 4.66 |
| TreeRep | 3.44 | 3.90 | 6.03 | 4.96 | 7.92 | 8.49 | 8.09 | 8.74 | 17.23 |
| Poincaré | _35.73_ | _34.94_ | _53.10_ | _35.24_ | _34.45_ | 52.71 | 34.47 | 33.70 | 52.19 |
| Hyperboloid | 35.56 | 34.77 | 51.90 | 34.80 | 34.01 | _52.88_ | _34.93_ | _34.15_ | _52.98_ |
| UnitBall | **44.05** | **43.26** | **61.54** | **47.88** | **46.96** | **70.28** | **46.54** | **45.59** | **70.03** |
| | WordNet-noun | | | | | | | | |
| | 8-dimensional | | | 32-dimensional | | | 128-dimensional | | |
| | MAP | MRR | Hits@3 | MAP | MRR | Hits@3 | MAP | MRR | Hits@3 |
| Euclidean | 1.07 | 1.05 | 0.63 | 5.59 | 5.36 | 3.16 | 14.33 | 13.35 | 8.82 |
| Poincaré | _25.23_ | _23.78_ | 27.63 | 25.46 | 23.99 | _27.80_ | 25.33 | 23.86 | 27.41 |
| Hyperboloid | **25.73** | **24.24** | _27.67_ | _25.65_ | _24.15_ | 27.50 | _25.77_ | _24.27_ | _27.65_ |
| UnitBall | 24.91 | 23.76 | **30.27** | **27.29** | **25.93** | **32.95** | **27.29** | **25.91** | **32.77** |

Table 8: Results of MAP and Hits@1 on predicting 1-$N$ edges in YAGO3-wikiObjects. The embedding dimension is 16 for UnitBall while 32 for other models. The best results are shown in boldface.

| $N$ for 1-$N$ edges | YAGO3-wikiObjects (MAP) | | | | | |
| | 1 | $2 \sim 5$ | $6 \sim 10$ | $11 \sim 20$ | $> 20$ | $> 1$ |
|---|---|---|---|---|---|---|
| Poincaré | 60.73 | 16.87 | 9.28 | 9.31 | 9.46 | 15.29 |
| Hyperboloid | **61.49** | 15.01 | 9.25 | 9.50 | 9.54 | 13.80 |
| UnitBall | 58.41 | **26.73** | **13.28** | **10.17** | **11.64** | **23.61** |
| | YAGO3-wikiObjects (Hits@1) | | | | | |
| Poincaré | 36.94 | 12.13 | 0.97 | 0.87 | 0.00 | 9.83 |
| Hyperboloid | **38.24** | 12.20 | 0.97 | 0.95 | 0.00 | 6.95 |
| UnitBall | 37.64 | **28.63** | **18.98** | **11.82** | **17.73** | **26.02** |

improvements, but its performances in 128-d still cannot surpass UnitBall and the hyperbolic models in 8-d. UnitBall outperforms the baselines almost all the time. Although on WordNet-noun, UnitBall in 4-d has slightly lower MAP and MRR than Poincaré and Hyperboloid in 8-d, it has much higher Hits@3.

## F.8   MORE RESULTS ON 1-$N$ STRUCTURE

In Section 5.3.3, we explored the performances on 1-$N$ structures. Here we present results of more evaluation metrics in Table 8. The results of UnitBall are close with Poincaré and Hyperboloid in 1-1 structure. Nevertheless, UnitBall outperforms the hyperbolic baselines by a big margin in 1-$N$ structure for $N > 1$. Moreover, Unitball has considerably huge improvement for $N > 6$, where the hyperbolic embedding models fail to make accurate predictions.

## F.9   COMPARISON WITH TRAINABLE CURVATURE METHOD ATTH

Our work focuses on the representation of single-relation graphs, which is a different research topic with multi-relational graph embeddings or knowledge graph embeddings, so it is hard to find an appropriate experimental setting to compare them. Nevertheless, to address the concerns of comparison with the trainable curvature method, here we evaluate AttH (Chami et al., 2020) on the single-relation taxonomy link prediction task. We tune the hyperparameters on the validation set and report the mean results over 5 running executions.

Table 9: Evaluation of taxonomy link prediction on YAGO3-wikiObjects (the dimension is 32 for AttH and 16 for UnitBall).

|          | MAP   | MRR   | Hits@1 | Hits@3 |
|----------|-------|-------|--------|--------|
| AttH     | 30.22 | 28.47 | 9.10   | 43.83  |
| UnitBall | 33.33 | 31.85 | 15.62  | 47.41  |

Table 10: Evaluation on link prediction task of GIL paper in ROC AUC (the dimension is 8 for UnitBall and 16 for HGCN and GIL).

|          | Disease | Airport | Pubmed | Citeseer | Cora  |
|----------|---------|---------|--------|----------|-------|
| HGCN     | 90.80   | 96.43   | 95.13  | 96.63    | 93.81 |
| GIL      | 99.90   | 98.77   | 95.49  | 99.85    | 98.28 |
| UnitBall | 99.09   | 96.61   | 98.80  | 99.34    | 97.64 |

From the results in Table 9, we see that UnitBall outperforms AttH in the single hypernymy relation link prediction task. However, UnitBall cannot infer multiple relations like AttH for now. MWe believe the future work of the complex hyperbolic embeddings will have promising improvements on multi-relational graph embeddings.

### F.10 COMPARISON WITH HYPERBOLIC GNNs

Our work focuses on the single-relation graph embeddings and taxonomy embeddings, so we do not evaluate the neural networks in our tasks in the main body of the paper. Although hyperbolic GNNs also involve graph embeddings and can deal with the link prediction task, they make use of not only the edges between nodes but also the node features. The message propagation and attention mechanism make GNNs more flexible to handle various downstream tasks than shallow embeddings. In this section, we evaluate UnitBall on the link prediction task on the datasets of GIL (Zhu et al., 2020). The results of HGCN (Chami et al., 2019) and GIL (Zhu et al., 2020) are copied from Table 2 of the GIL's original paper (Zhu et al., 2020). We strictly follow their experimental settings and report the mean results of UnitBall in ROC AUC over 5 running executions. The dimension is 8 for UnitBall and 16 for HGCN and GIL.

We can see that UnitBall outperforms HGCN on the five datasets. GIL is slightly better than UnitBall on most datasets while being outperformed by UnitBall on Pubmed. The results are very promising for UnitBall since UnitBall is a shallow embedding approach without deep architecture or feature interaction. We believe the complex hyperbolic embeddings will help to improve the GNNs and bring more insights into geometric deep learning.

### F.11 COMPARISON WITH PRODUCT EMBEDDINGS

As mentioned in Section 2, the product space embeddings (Gu et al., 2019) tackles the challenges in varying local structures by jointly learning the curvature and the embeddings of data in a product manifold. Although it is impractical to search for the best manifold combination among enormous combinations for each new structure, it is worth exploring the comparisons between the complex hyperbolic embeddings and products of hyperbolic embeddings. Therefore, in this section, we conduct experiments on the reconstruction task in synthetic compressed graphs. We evaluated the 16-dimensional UnitBall complex hyperbolic embeddings and 32-dimensional product hyperbolic embeddings on $(\mathbb{H}_{\mathbb{R}}^2)^{16}, (\mathbb{H}_{\mathbb{R}}^4)^8, (\mathbb{H}_{\mathbb{R}}^8)^4$. The results are reported in Table 11.

Recall that each compressed graph-$(m, k)$ consists of $m$ nodes and is aggregated from $k$ random trees on the $m$ nodes. The bigger $k$ corresponds to the denser and noisier graph. When $k = 1$ ($\delta = 0$), the graph is exactly a tree structure, UnitBall and the product hyperbolic embeddings both have much better performances in this case. When $k > 1$, UnitBall still outperforms the product hyperbolic embeddings by a large margin. Especially when $m = 500, k = 2, 3$, the $\delta$ is big, which means the graph deviates from tree structures a lot, the product hyperbolic embeddings fail to reconstruct the graph while UnitBall successfully handles the noisy structures.

Table 11: Results of MAP and Hits@3 on graph reconstruction in synthetic compressed graphs. The best results are shown in boldface. The second best results are underlined.

| | Compressed graph-$(m,k)$ (MAP) | | | | | | | | | |
|---|---|---|---|---|---|---|---|---|---|---|
| k (m=500) | 1 | 2 | 3 | 4 | 5 | 6 | 7 | 8 | 9 | 10 |
| $\delta$-hyperbolicity | 0.0 | 2.5 | 1.5 | 1.0 | 1.0 | 1.0 | 1.0 | 1.0 | 1.0 | 1.0 |
| Product-$(\mathbb{H}_{\mathbb{R}}^2)^{16}$ | 66.06 | 7.60 | 7.55 | 11.09 | 20.81 | 24.56 | 24.07 | 21.62 | 18.79 | 16.16 |
| Product-$(\mathbb{H}_{\mathbb{R}}^4)^{8}$ | 65.77 | 7.14 | 7.24 | 11.79 | 20.60 | 24.94 | 22.85 | 22.32 | 18.50 | 16.27 |
| Product-$(\mathbb{H}_{\mathbb{R}}^8)^{4}$ | 65.42 | 6.28 | 6.81 | 11.43 | 19.03 | 23.88 | 24.99 | 20.50 | 18.99 | 16.45 |
| UnitBall-$\mathbb{H}_{\mathbb{C}}^{16}$ | **84.72** | **52.74** | **44.73** | **39.75** | **35.32** | **33.17** | **32.48** | **29.13** | **29.86** | **28.58** |
| | Compressed graph-$(m,k)$ (Hits@3) | | | | | | | | | |
| Product-$(\mathbb{H}_{\mathbb{R}}^2)^{16}$ | 80.34 | 8.14 | 9.84 | 14.81 | 37.45 | 45.97 | 47.89 | 47.19 | 40.85 | 38.88 |
| Product-$(\mathbb{H}_{\mathbb{R}}^4)^{8}$ | 80.34 | 8.35 | 9.43 | 16.23 | 36.23 | 47.98 | 44.87 | 47.19 | 39.84 | 36.87 |
| Product-$(\mathbb{H}_{\mathbb{R}}^8)^{4}$ | 79.36 | 6.21 | 10.25 | 16.63 | 31.38 | 44.15 | 49.50 | 43.57 | 40.64 | 40.08 |
| UnitBall-$\mathbb{H}_{\mathbb{C}}^{16}$ | **97.71** | **75.87** | **72.61** | **74.92** | **73.21** | **73.12** | **76.12** | **76.57** | **75.72** | **74.08** |
| | Compressed graph-$(m,k)$ (MAP) | | | | | | | | | |
| m (k=5) | 100 | 200 | 300 | 400 | 500 | 600 | 700 | 800 | 900 | 1000 |
| $\delta$-hyperbolicity | 1.0 | 1.0 | 1.0 | 1.0 | 1.0 | 1.5 | 1.5 | 1.5 | 1.5 | 1.5 |
| Product-$(\mathbb{H}_{\mathbb{R}}^2)^{16}$ | 30.53 | 29.15 | 25.72 | 21.74 | 20.81 | 19.34 | 19.07 | 16.63 | 15.59 | 14.82 |
| Product-$(\mathbb{H}_{\mathbb{R}}^4)^{8}$ | 32.03 | 27.63 | 23.82 | 22.32 | 20.60 | 17.94 | 19.12 | 17.36 | 15.39 | 14.74 |
| Product-$(\mathbb{H}_{\mathbb{R}}^8)^{4}$ | 31.39 | 27.95 | 23.17 | 19.90 | 19.03 | 17.96 | 18.09 | 16.13 | 13.86 | 13.47 |
| UnitBall-$\mathbb{H}_{\mathbb{C}}^{16}$ | **47.80** | **40.93** | **38.52** | **35.81** | **35.32** | **35.68** | **34.73** | **34.69** | **34.92** | **34.88** |
| | Compressed graph-$(m,k)$ (Hits@3) | | | | | | | | | |
| Product-$(\mathbb{H}_{\mathbb{R}}^2)^{16}$ | 38.78 | 52.02 | 43.58 | 35.61 | 37.45 | 34.29 | 34.54 | 28.34 | 27.44 | 24.77 |
| Product-$(\mathbb{H}_{\mathbb{R}}^4)^{8}$ | 46.94 | 43.43 | 39.53 | 38.38 | 36.23 | 29.58 | 35.84 | 29.35 | 27.55 | 27.60 |
| Product-$(\mathbb{H}_{\mathbb{R}}^8)^{4}$ | 54.08 | 42.42 | 35.47 | 33.59 | 31.38 | 32.44 | 30.49 | 29.72 | 24.64 | 24.57 |
| UnitBall-$\mathbb{H}_{\mathbb{C}}^{16}$ | **84.35** | **75.42** | **77.36** | **72.14** | **73.21** | **73.00** | **73.99** | **72.63** | **72.23** | **74.01** |

Table 12: Nodes embedded in the most simple totally real plane of the unit ball model. The embeddings are trained by UnitBall model on link prediction task in Section 5.3.1. Considering the possible numerical error, we allow the embeddings to deviate the plane for a small threshold instead of strictly lying in the plane.

| Datasets | Nodes embedded in the totally real plane |
|---|---|
| ICD10 | ICD10-root |
| YAGO3-wikiObjects | Objects (root), Physical objects, Organisms, Men |
| WordNet-noun | entity.n.01 (root) |

## F.12    CASE STUDY OF NODES EMBEDDED IN THE TOTALLY REAL PLANE

Although as analyzed in Appendix B, curvature in the complex hyperbolic space is a very complicated topic in geometric group theory and differential geometry. Computing the curvature of an arbitrary point in the complex hyperbolic space or visualizing the curvatures everywhere are highly complicated problems, the existing work usually explored a few examples by numerical computation. However, the curvatures in two special subspaces, i.e., the complex projective lines and the totally real planes, are well-studied. It is interesting to see which nodes lie in the special subspaces. In this section, we find the nodes of the real-world taxonomies that are embedded in the most simple totally real plane of the unit ball model, i.e., $\mathbb{H}_{\mathbb{R}}^n \cap \mathcal{B}_{\mathbb{C}}^n$, where $\mathbb{H}_{\mathbb{R}}^n = \{(x_1, \ldots, x_n)|x_1, \ldots, x_n \in \mathbb{R}\}$.

Table 12 shows that the root node of all three taxonomies are embedded in the totally real plane, which is in accordance with expectations since the root nodes are usually embedded close with the origin. For the more fine-grained taxonomy YAGO3-wikiObjects, some high-level nodes are also embedded in this totally real plane. These nodes actually form a chain of hypernymy relation in the taxonomy: $Men \rightarrow Organisms \rightarrow Physical\ objects \rightarrow Objects$. Recall Theorem 1 states

that the curvature in complex hyperbolic space is pinched between $-1$ (in the directions of complex projective lines) and $-1/4$ (in the directions of totally real planes). From the results in the three taxonomies, we can see that the high-level nodes in the hierarchical structure tend to lie in the totally real planes, which are the least curved subspaces of the complex hyperbolic space.

