# OpenReview forum: "Unit Ball Model for Embedding Hierarchical Structures in the Complex Hyperbolic Space"
_ICLR.cc/2022/Conference — ICLR 2022 Submitted_

### Official Review · Reviewer_N6rj · 2021-10-26

**Correctness:** 3
**Technical Novelty And Significance:** 3
**Empirical Novelty And Significance:** 3
**Recommendation:** 6
**Confidence:** 4

**Main Review:**

## Pros:

(+) The idea and intuition of introducing complex hyperbolic embedding are novel and clear.

(+) The experiment shows the great performance of the proposed method.

## Cons:

(-) Although it is intuitively correct that variable curvature can better suit the real-world data, it is not well demonstrated in the paper besides the performance metric in the experiment. I hope there will be a figure to illustrate at least one such example.

(-) In the experiment section, the authors only compare with the standard (real) hyperbolic embedding method. Why not compare with the product space with multiple hyperbolic signatures?

## Detail comments

Overall, I like the paper where their key idea is interesting and inspiring. The extensive experiments on different tasks and datasets (including both synthetic and real-world) are appreciated. However, I have some slight concerns about the paper.

First, I get that intuitively we want variable curvature spaces to account for varying local structures in data. However, I wonder what is the exact effect of such “variable curvature”? For example, is it possible to show (either by figure or experiment) that certain graph structures induced embeddings concentrate more at the place with smaller/larger curvature? I think one such example can greatly improve the overall clarity, as we do see a direct effect of variable curvature in the complex hyperbolic space.

Second, the authors keep mentioning that their embedding is different from the product space of two n-dimensional real hyperbolic spaces. While I can get this through the analysis provided by the authors, it is then natural to ask whether it is better or not. I find that the authors do not conduct such experiments and thus I think it is unsatisfactory. For example, given dimension 8, can the 4-dim UnitBall embedding outperform embeddings from product spaces of 4H^2 and 2H^4? Note that according to the experiments in [1], these products of small dimension hyperbolic spaces give much better results compared to H^8 (single large dimension hyperbolic space). See also the similar findings in terms of classification downstream tasks in [2]. Also, I do not fully understand why the authors do not compare UnitBall with neural network based embedding methods such as mixed-curvature VAE [3]. Even if restricting the signature to be hyperbolic is good enough. Please either provide more experiments or elaborate on why such an experiment is not done.

Finally, I find the related work section can be slightly improved. Although neural networks are overwhelming nowadays in hyperbolic learning. Classical methods such as perceptrons and SVM [4,5] should not be ignored, as they come with convergence guarantees.

## References

[1] LEARNING MIXED-CURVATURE REPRESENTATIONS IN PRODUCTS OF MODEL SPACES, Gu et al. ICLR 2019.

[2] Linear Classifiers in Product Space Forms, Tabaghi et al. ArXiv 2021.

[3] Mixed-curvature variational ´ autoencoders, Skopek et al. ICLR 2020.

[4] Large-margin classification in hyperbolic space, Cho et al. AISTATS 2019.

[5] Highly Scalable and Provably Accurate Classification in Poincare Balls, Chien et al. ICDM 2021.


**Summary Of The Paper:**

The authors propose to learn complex hyperbolic embedding, which outperforms the real hyperbolic embedding in graph reconstruction and link prediction tasks. The authors argue that the complex hyperbolic space can have variable negative curvature, which can better capture the varying local structures in real-world data.

**Summary Of The Review:**

Despite the slight concerns mentioned above, I still think the idea and the methodology of the paper is good enough for ICLR. The proposed complex hyperbolic spaces can potentially inspire more advanced method in the field. Hence, I lean to recommend accept of this paper.

---

> ### Author Response · Authors · 2021-11-16
> **Response to Reviewer N6rj (Part 1: Experiments on Curvature )**
>
> We would like to thank the reviewer for your constructive comments and suggestions, as well as your appreciation of our work. First, we want to emphasize that our task is the representation learning of hierarchically structured graph data, specifically, the real-world taxonomies and multitrees. Accordingly, our experimental tasks focus on graph reconstruction and link prediction instead of classification or image reconstruction. Next, we will address your detailed comments as follows.
>
> > I wonder what is the exact effect of such “variable curvature”? For example, is it possible to show (either by figure or experiment) that certain graph structures induced embeddings concentrate more at the place with smaller/larger curvature? I think one such example can greatly improve the overall clarity, as we do see a direct effect of variable curvature in the complex hyperbolic space.
>
> We agree that a figure to visualize the embeddings and curvatures would make the understanding easier and much clearer. However, curvature in the complex hyperbolic space is a very complicated topic in geometric group theory and differential geometry. As far as we know, there is no prior work or research study that computes the curvature of an arbitrary point in the complex hyperbolic space or visualizes the curvatures everywhere. As you can see in Appendix B, the curvatures of the two special subspaces, i.e., the complex projective lines and the totally real planes, are obtained by mapping their metrics with the well-studied model metrics (the Poincar\'e metric and the Klein-Beltrami metric). For curvatures in other subspaces, the existing work usually explored a few examples by numerical computation (Fisher). The visualization of the complex hyperbolic embeddings and curvatures calls for deeper development of the complex hyperbolic geometry research as well as the visualization study of geometric models. We also eagerly expect the related work to come out.
>
> Inspired by your suggestion, we add extra experiments to find the nodes whose embeddings lie in the most simple totally real plane of the unit ball model, i.e., $\mathbb{H}_\mathbb{R}^n\cap\mathcal{B}_\mathbb{C}^n$, where $\mathbb{H}_\mathbb{R}^n=\{(x_1,\dots,x_n)|x_1,\dots,x_n\in\mathbb{R}\}$. As proved in Appendix B, the curvature of the totally real planes is $-1/4$. We use the embeddings trained by UnitBall on link prediction task in Section 5.3.1. We get some interesting findings.
>
> | Datasets          | Nodes embedded in the totally real plane         |
> |-------------------|--------------------------------------------------|
> | ICD10             | ICD10-root                                       |
> | YAGO3-wikiObjects | Objects (root), Physical objects, Organisms, Men |
> | WordNet-noun      | entity.n.01 (root)                               |
>
> The table shows that the root node of all three taxonomies are embedded in the totally real plane, which is in accordance with expectations since the root nodes are usually embedded close with the origin. For the more fine-grained taxonomy YAGO3-wikiObjects, some high-level nodes are also embedded in the totally real plane. These nodes actually form a chain of hypernymy relation in the taxonomy: $Men \to Organisms \to Physical \text{ } objects \to Objects$. Recall Theorem 1 states that the curvature in complex hyperbolic space is pinched between $-1$ (in the directions of complex projective lines) and $-1/4$ (in the directions of totally real planes). From the results in the three taxonomies, we can see that the high-level nodes in the hierarchical structure tend to lie in the totally real planes, which are the least curved subspaces of the complex hyperbolic space. We add the above results in Appendix F.12 in our updated revision.
>
> **References**:
>
> N. Fisher. Notes on curvature in complex hyperbolic space. URL https://sites.tufts. edu/natefisher/files/2020/11/Write-up.pdf.

---

> ### Author Response · Authors · 2021-11-16
> **Response to Reviewer N6rj (Part 2:  Comparison with Product Embeddings and NN-Based Embeddings )**
>
> > The authors keep mentioning that their embedding is different from the product space of two n-dimensional real hyperbolic spaces. While I can get this through the analysis provided by the authors, it is then natural to ask whether it is better or not. I find that the authors do not conduct such experiments and thus I think it is unsatisfactory. For example, given dimension 8, can the 4-dim UnitBall embedding outperform embeddings from product spaces of $4H^2$ and $2H^4$?
>
> Although it is impractical to search for the best manifold combination among enormous combinations of the product spaces for each new structure, we agree with your comment that it is worth exploring the comparisons between the complex hyperbolic embeddings and products of hyperbolic embeddings. Therefore, we conduct experiments on the reconstruction task in synthetic compressed graphs. We evaluated the $16$-dimensional UnitBall complex hyperbolic embeddings and $32$-dimensional product hyperbolic embeddings (Gu et al., 2019) on $(\mathbb{H}_\mathbb{R}^2)^{16}$, $(\mathbb{H}_\mathbb{R}^4)^8$, $(\mathbb{H}_\mathbb{R}^8)^4$. The MAP results are reported in the following table.
>
> |  k ( m=500)                              | 1     | 2     | 3     | 4     | 5     | 6     | 7     | 8     | 9     | 10    |
> |------------------------------------------|-------|-------|-------|-------|-------|-------|-------|-------|-------|-------|
> | $\delta$-hyperbolicity                   | 0.0   | 2.5   | 1.5   | 1.0   | 1.0   | 1.0   | 1.0   | 1.0   | 1.0   | 1.0   |
> | Product-$(\mathbb{H}_\mathbb{R}^2)^{16}$ | 66.06 | 7.60  | 7.55  | 11.09 | 20.81 | 24.56 | 24.07 | 21.62 | 18.79 | 16.16 |
> | Product-$(\mathbb{H}_\mathbb{R}^4)^8$    | 65.77 | 7.14  | 7.24  | 11.79 | 20.60 | 24.94 | 22.85 | 22.32 | 18.50 | 16.27 |
> | Product-$(\mathbb{H}_\mathbb{R}^8)^4$    | 65.42 | 6.28  | 6.81  | 11.43 | 19.03 | 23.88 | 24.99 | 20.50 | 18.99 | 16.45 |
> | UnitBall-$\mathbb{H}_\mathbb{C}^{16}$    | **84.72** | **52.74** | **44.73** | **39.75** | **35.32** | **33.17** | **32.48** | **29.13** | **29.86** | **28.58** |
>
> Recall that each compressed graph-$(m, k)$ consists of $m$ nodes and is aggregated from $k$ random trees on the $m$ nodes. The bigger $k$ corresponds to the denser and noisier graph. We also give the $\delta$-hyperbolicty of the graphs in the table. When $k=1$ ($\delta=0$), the graph is exactly a tree structure, UnitBall and the product hyperbolic embeddings both have much better performances in this case. When $k>1$, UnitBall still outperforms the product hyperbolic embeddings by a large margin. Especially when $k=2, 3$, the $\delta$ is big, which means the graph deviates from tree structures a lot, the product hyperbolic embeddings fail to reconstruct the graph while UnitBall successfully handles the noisy structures. We add the experiment in Appendix F.11 in our updated revision, where you can check for more results.
>
> > I do not fully understand why the authors do not compare UnitBall with neural network based embedding methods such as mixed-curvature VAE [3]. Even if restricting the signature to be hyperbolic is good enough. Please either provide more experiments or elaborate on why such an experiment is not done.
>
> We do compare UnitBall with neural network based embedding methods. The results are reported in Appendix F.10, where we compare with the mix-curvature hyperbolic graph neural network HGCN (Chami et al., 2019) and graph geometry interaction learning method GIL (Zhu et al., 2020). The results show that UnitBall outperforms HGCN on the five datasets. GIL is slightly better than UnitBall on most datasets while being outperformed by UnitBall on Pubmed. The results are very promising for UnitBall since UnitBall is a shallow embedding approach without deep architecture or feature interaction. We believe the complex hyperbolic embeddings will help to improve the GNNs and bring more insights into geometric deep learning. For more details, please refer to Appendix F.10.
>
> As for the mixed-curvature VAE (Skopek et al., 2020), the work focuses on modeling the probability distributions in product spaces and evaluates on the image reconstruction task. Both the methodology (VAE framework for modeling distributions) and the experimental settings (e.g., tasks, baselines, types of datasets) are in different research scope with our work.
>
> **References**:
>
> I. Chami, Z. Ying, C. Re ́, and J. Leskovec. Hyperbolic graph convolutional neural networks. In NeurIPS, pages 4869–4880, 2019.
>
> A. Gu, F. Sala, B. Gunel, and C. Re ́. Learning mixed-curvature representations in product spaces. In ICLR (Poster). OpenReview.net, 2019.
>
> O. Skopek, O. Ganea, and G. Be ́cigneul. Mixed-curvature variational autoencoders. In ICLR. OpenReview.net, 2020.
>
> S. Zhu, S. Pan, C. Zhou, J. Wu, Y. Cao, and B. Wang. Graph geometry interaction learning. In NeurIPS, 2020.

---

> ### Author Response · Authors · 2021-11-16
> **Response to Reviewer N6rj (Part 3: Minor Comments on Related Work)**
>
> > Finally, I find the related work section can be slightly improved. Although neural networks are overwhelming nowadays in hyperbolic learning. Classical methods such as perceptrons and SVM [4,5] should not be ignored, as they come with convergence guarantees.
>
> Thanks for the suggestion. Due to the page limit, we only introduce the related works which study the representation learning of hierarchical graphs and focus on graph reconstruction as well as link prediction task. In our updated revision, we add works on other research tasks and applications inspired by hyperbolic learning in Related Work.
>
> We would like to thank you again for your efforts to improve our paper. Please let us know if you have any further concerns or questions. We look forward to having more discussions with you.

---

> ### Comment · Reviewer_N6rj · 2021-11-29
> **Re**
>
> Thanks for the response. After reading the response and the other reviews, I still think it is an interesting work. Nevertheless, I also agree with some of the other reviewer's concerns. Hence, I keep my score unchanged. I hope the authors can add the new results in the response to their revision, especially to further clarify the motivation of using complex embedding.

---

> > ### Author Response · Authors · 2021-11-29
> > **Thanks for your reply**
> >
> > Thank you for your reply. We have added the new results in our rebuttal revision. Please see our comment *Many thanks to all reviewers and Summary of the revisions* and check our updated revision for details. Thanks again for your constructive suggestions and your appreciation of our work!

---

### Official Review · Reviewer_ZxKu · 2021-11-02

**Correctness:** 4
**Technical Novelty And Significance:** 2
**Empirical Novelty And Significance:** 2
**Recommendation:** 5
**Confidence:** 4

**Main Review:**

## Strengths
 The unit ball model-based embeddings have a more powerful representation capacity to capture a variety of hierarchical structures. Through experiments on synthetic and real-world data, the author's work shows promising empirical results.

## Weakness
**1. The motivation is not clear to me. How do complex hyperbolic spaces solve the varying local structures?**

The network structure in the real world is complicated, and the use of one space with a constant curvature is likely to cause distortion. However,  the curvature of hyperbolic space is global. How does it deal with different local structures in the same network? For example, there are both tree local structures and ring local structures in a network. When using a global curvature -1, it is still difficult to take into account the two local structures simultaneously.

**2. What kind of graphs are suitable to be embedded in the complex hyperbolic space is not clear**

(More questions could be raised in the following discussion.)



**Summary Of The Paper:**

In most real-world hierarchically structured data such as taxonomies and multitree networks have varying local structures and they
are not trees, thus they do not ubiquitously match the constant curvature property of the hyperbolic space. The authors explore the complex hyperbolic space when EMBEDDING HIERARCHICAL STRUCTURES.

**Summary Of The Review:**

The unit ball model-based embeddings have a more powerful representation capacity to capture a variety of hierarchical structures. Through experiments on synthetic and real-world data, the author's work shows promising empirical results. However, there are two key problems causing my concerns.

---

> ### Author Response · Authors · 2021-11-16
> **Response to Reviewer ZxKu**
>
> We would like to thank the reviewer for your valuable questions and for affirming our promising empirical results. In the following, we will address your questions.
>
> > The motivation is not clear to me. How do complex hyperbolic spaces solve the varying local structures?
>
> We want to emphasize that the complex hyperbolic geometry differs from the real hyperbolic geometry intrinsically. The complex hyperbolic space has **variable** negative curvature, which is addressed in Abstract, Introduction, and formally presented in Theorem 1. We totally agree with your statement that the hyperbolic space with global constant curvature is difficult to handle the complicated structures of the real-world data. That is our motivation to learn the embeddings in the complex hyperbolic space, i.e., to make the **non-constant** curvature provide the capacity for more varying structures.
>
> > What kind of graphs are suitable to be embedded in the complex hyperbolic space is not clear.
>
> As supported by the theoretical guarantees in Section 3.2, the variable negative curvature of the complex hyperbolic space is expected to be favorable for embedding various hierarchical structures while the tree-like exponential volume growth property maintains the advantages in tree structures. Our experiments demonstrated our claims that the complex hyperbolic embeddings handle **various hierarchically structured data** better, including **synthetic balanced trees** (Appendix F.5), **synthetic compressed graphs** (Section 5.2.1), and **real-world taxonomies** (Section 5.3). In addition, we investigates two specific structures where complex hyperbolic geometry shows outstanding performances, namely the **multitree structure** (Section 5.2.2) and **1-$N$ structure** (Section 5.3.3).
>
> We would like to thank you again for your efforts to review our paper. We hope our response can clarify your confusion. More questions are welcome. We look forward to having further discussions with you.

---

### Official Review · Reviewer_PCHz · 2021-11-02

**Correctness:** 4
**Technical Novelty And Significance:** 2
**Empirical Novelty And Significance:** 2
**Recommendation:** 5
**Confidence:** 4

**Main Review:**

The paper is well written in general, and the optimization framework is clear although not novel since it follows standard Riemannian optimization.

- One major weakness of the paper is that the motivation of using the complex hyperbolic space instead of more recent approaches such as mixed curvature representations [E] or ultrahyperbolic embeddings [F] is not clear. In Section 2, the authors argue that it is impractical to search for the best manifold combination of [E] for each graph. Therefore, ref [E] is ignored from the baselines in the experimental section and the only relevant baselines are standard hyperbolic approaches [B,C] that are known to be outperformed by most of their variants. The improvement over baselines [B,C] does not seem significant so it is difficult to justify using the complex hyperbolic space. At least, ref [E] should have been used for comparison.

Moreover, even after reading the appendix and the experimental section, I do not understand the type of relationship that the distance function in Eq. (9) is supposed to describe. In [B,C,E,F], the motivation of the geodesic distances is given (e.g., high-level nodes tend to be closer to the origin in [B,C]). In the submission, there is no illustration or toy dataset/analysis to understand the embeddings that are learned.

Most of the details (e.g., description of the dataset, optimization framework) could have been moved to the appendix to give a better intuition of the learned representations to the reader.

- Another weakness of the paper is the fact that the learned representations are nonparametric embeddings. For instance, hyperbolic embeddings have been extended to hyperbolic (graph) neural networks, just like ref [E] was extended to graph neural networks [G]. How easy is it to extend the exploited manifold to (graph) neural network and perform inference of test examples?

[A] Goldman, Complex Hyperbolic Geometry, Oxford University Press, 1999

[B] Nickel and Kiela, Poincaré embeddings for learning hierarchical representations, NIPS 2017

[C] Nickel and Kiela, Learning continuous hierarchies in the Lorentz model of hyperbolic geometry, ICML 2018

[D] Absil et al, Optimization algorithms on matrix manifolds. Princeton University Press, 2009

[E] Gu et al., Learning Mixed-Curvature representations in product spaces, ICLR 2019

[F] Law and Stam, Ultrahyperbolic representation learning, NeurIPS 2020

[G] Bachmann et al., Constant curvature graph convolutional networks, ICML 2020

**Summary Of The Paper:**

The paper introduces an extension of real hyperbolic embeddings to the complex hyperbolic space [A]. The exploited geometry is an extension of the Poincaré ball that contains complex vectors (instead of real vectors) whose norm is smaller than 1. The resulting manifold is of nonconstant negative curvature, which the authors expect to be favorable for embedding various hierarchical structures.
Following the optimization framework of [B] and since the complex hyperbolic space is a Riemannian manifold, Section 4.3 presents a standard Riemannian optimization framework to learn nonparametric embeddings. The proposed manifold shows (slightly) superior results compared to real hyperbolic embeddings proposed in [B,C] in the graph reconstruction and link prediction tasks.

**Summary Of The Review:**

The idea looks interesting but the paper lacks providing the intuition of the learned representations, and baselines are lacking.

---

> ### Author Response · Authors · 2021-11-16
> **Response to Reviewer PCHz (Part 1: Motivation)**
>
> We would like to thank the reviewer for your constructive comments and your appreciation of our writing. First, we want to emphasize that one of the main contributions of our paper is the proposition of the novel embedding approach defined in the unit ball model of complex hyperbolic space to handle nosier and more flexible hierarchies. Neither of the complex hyperbolic embeddings or the complex RSGD formulation has been defined or proposed before. Next, we will address your questions and comments as follows.
>
> > One major weakness of the paper is that the motivation of using the complex hyperbolic space instead of more recent approaches such as mixed curvature representations [E] or ultrahyperbolic embeddings [F] is not clear.
>
> The motivation for using the complex hyperbolic space comes from its favorable geometric properties. Specifically, the variable negative curvature of the complex hyperbolic geometry can address the limitation of hyperbolic embeddings in real-world hierarchically structured data since the varying local structures do not ubiquitously match the constant curvature property of the hyperbolic space. In addition, the complex hyperbolic geometry still maintains the tree-like exponential volume growth property. Therefore, the complex hyperbolic space has a more powerful representation capacity for embedding hierarchical structures.
>
> Besides the enormous search space issue mentioned in Related Work, the mixed curvature representations (Gu et al., 2019), as a combinatorial construction-based embedding method, also suffered from unpromising generalization performance in the link prediction task (addressed in Section 5.3.1) and the memory issue (addressed in Appendix F.6). Please note that our experiments evaluated the SOTA construction-based hyperbolic embedding method TreeRep (Sonthalia and Gilbert, 2020), while the aforementioned two issues are generally shared by construction-based methods. As for the ultrahyperbolic embeddings (Law and Stam, 2020), it proposed a representation defined on a pseudo-Riemannian manifold which is a generalization of hyperbolic and spherical geometries, but the proposed manifold still has constant curvature.
>
> We sincerely appreciate the contributions of previous works, including the above two works mentioned by the reviewer. There are different research lines in the exciting hyperbolic embeddings field, where each methodology tackles some focused challenges while inspiring and improving each other. We aimed at tackling the limitations of previous works on complicated hierarchical structures deviating from tree metrics. The proposed complex hyperbolic embedding has not been explored in graph representation learning before. We analyzed its related geometric properties, formulated the novel learning framework, and demonstrated its advantages through evaluating on an extensive range of synthetic and real-world data.
>
> **References**:
>
> A. Gu, F. Sala, B. Gunel, and C. Re ́. Learning mixed-curvature representations in product spaces. In ICLR (Poster). OpenReview.net, 2019.
>
> M. T. Law and J. Stam. Ultrahyperbolic representation learning. InNeurIPS, 2020.
>
> R. Sonthalia and A. C. Gilbert. Tree! I am no tree! I am a low dimensional hyperbolic embedding. In NeurIPS, 2020.

---

> > ### Comment · Reviewer_PCHz · 2021-11-27
> > **Thank you for your explanation.**
> >
> > After reading the explanation, updated Appendix A.2 and the referred Chapter 3.1 of (Goldman, 1999), I understand that the proposed manifold is a generalization of the Poincaré ball to the complex domain where, instead of having constant negative sectional curvature, the manifold has constant holomorphic sectional curvature equal to -1 (see page 76 of (Goldman, 1999)). Chapter 3.1 of (Goldman, 1999) is a great reference to understand the properties of the manifold but it is still difficult to understand what kinds of relationships between nodes are described.
> >
> > As explained by the authors, the fact the manifold has different local curvatures in the interval [-1,-1/4] is supposed to allow the manifold to describe different kinds of tree-like structures. Is there such kind of analysis in the paper/appendix? I see in the last paragraph of Appendix F.12 that high-level nodes tend to lie in the least curved parts of the manifold (i.e. where the curvature is close to -1/4). Why is it the case? Does it depend on the topology/properties of the graph? What kinds of graphs are better represented with complex hyperbolic geometry than real hyperbolic geometry? Since high-level nodes and low-level nodes live in regions with different curvature, how does it changes their distances wrt other nodes? I would assume that it means high-level nodes tend to be closer to other nodes than low-level nodes (i.e. their distance grows less exponentially).
> >
> > I believe that such an analysis is still lacking.

---

> > > ### Author Response · Authors · 2021-11-28
> > > **Thanks for your reply and further questions**
> > >
> > > We appreciate your reply and further questions very much. In the following we will address your new comments.
> > >
> > > Firstly, please note that **the unit ball model is not simply a generalization of the Poincaré ball model to the complex domain**. As referred to (Apanasov, 1997), the geometric properties of complex hyperbolic manifolds are surprisingly different from real hyperbolic manifolds. Secondly, we want to clarify that the holomorphic sectional curvature is a different concept from sectional curvature. **The complex hyperbolic space has constant holomorphic curvature as $-1$ while variable sectional curvature between $-1$ and $-1/4$** (see page 5 of (Apanasov, 1997)).
> > >
> > > Your questions about Appendix F.12 are very inspiring. We also think it is an interesting finding that the high-level nodes in the hierarchical structure tend to lie in the least curved subspaces. Your point that the less curved subspace has less exponentially growth is true. The results in Appendix F.12, especially the YAGO3-wikiObjects case can infer that hierarchy formed by high-level nodes tend to be embedded in the totally real plane. It is true that the high-level nodes tend to be closer with each other as well as with other lower-level nodes. The first reason comes from **their less-curved subspace** that you mentioned. The second reason is that **their embedding norms are generally smaller than low-level nodes** (as addressed in Appendix A.2, the distance function Eq. (9) maintains the tree-like metric properties, the point closer to origin has relatively smaller distances to the other points ). We will add the embedding norms of the high-level nodes in Appendix F.12 in our final version.
> > >
> > > We would like to emphasize that **the analyses of the complex hyperbolic geometry's properties in our paper are convincing and intuitively supportive for our claim** to address the limitations of the real hyperbolic space in real-world hierarchically structured data. Furthermore, **we empirically demonstrated the more powerful representation capacity of the complex hyperbolic geometry** on an extensive range of synthetic and real-world data.
> > >
> > > We agree that we did not provide proofs to strictly align the geometry structure and general graph metric, which would be a challenging and more theoretical topic in future work. We think the concern is insightful. However, it does not obscure the light of the contributions and importance of our work. Our contributions include not only **the novel embedding approach leveraging the complex hyperbolic geometry**, which has not been used in representation learning before, but also **the remarkable improvements on real-world taxonomies and some specific widely-used structures**. We believe our work can inspire more interesting future works in the aspect of **deeper theoretical explorations of complex hyperbolic representation learning** as well as **promising applications on multi-relational graph embeddings and neural networks**.
> > >
> > > We sincerely appreciate your time and insightful comments. Please let us know if you have further questions or suggestions to improve our paper.
> > >
> > > **References**:
> > >
> > > B. Apanasov.  Geometry and topology of complex hyperbolic and cr-manifolds.arXiv preprintmath/9701212, 1997.

---

> ### Author Response · Authors · 2021-11-16
> **Response to Reviewer PCHz (Part 2: Baselines and Results on Comparison with Product Embeddings)**
>
> > In Section 2, the authors argue that it is impractical to search for the best manifold combination of [E] for each graph. Therefore, ref [E] is ignored from the baselines in the experimental section and the only relevant baselines are standard hyperbolic approaches [B,C] that are known to be outperformed by most of their variants. The improvement over baselines [B,C] does not seem significant so it is difficult to justify using the complex hyperbolic space. At least, ref [E] should have been used for comparison.
>
> Please note that except for the Poincar ́e embeddings (Nickel and Kiela, 2017) and hyperboloid embeddings (Nickel and Kiela, 2018), our baseline also includes the SOTA construction-based hyperbolic method TreeRep (Sonthalia and Gilbert, 2020). In the graph reconstruction task on synthetic data (Section 5.2.1 \& Appendix F.5) and link prediction task on real-world data (Section 5.3), our method UnitBall achieves significant improvements over the baselines. Remarkably, UnitBall gained more than $12$ MAP points over the second-best method on ICD10 (Table 2) and more than $10$ Hits@10 points on 1-$N$ structure inference for $N>1$ (Table 4).
>
> As suggested by Reviewer N6rj, although it is impractical to search for the best manifold combination among enormous combinations of the product spaces for each new structure, it is worth exploring the comparisons between the complex hyperbolic embeddings and products of hyperbolic embeddings. Therefore, we conduct experiments on the reconstruction task in synthetic compressed graphs. We evaluated the $16$-dimensional UnitBall complex hyperbolic embeddings and $32$-dimensional product hyperbolic embeddings (Gu et al., 2019) on $(\mathbb{H}_\mathbb{R}^2)^{16}$, $(\mathbb{H}_\mathbb{R}^4)^8$, $(\mathbb{H}_\mathbb{R}^8)^4$. The MAP results are reported in the following table.
>
> |  k ( m=500)                              | 1     | 2     | 3     | 4     | 5     | 6     | 7     | 8     | 9     | 10    |
> |------------------------------------------|-------|-------|-------|-------|-------|-------|-------|-------|-------|-------|
> | $\delta$-hyperbolicity                   | 0.0   | 2.5   | 1.5   | 1.0   | 1.0   | 1.0   | 1.0   | 1.0   | 1.0   | 1.0   |
> | Product-$(\mathbb{H}_\mathbb{R}^2)^{16}$ | 66.06 | 7.60  | 7.55  | 11.09 | 20.81 | 24.56 | 24.07 | 21.62 | 18.79 | 16.16 |
> | Product-$(\mathbb{H}_\mathbb{R}^4)^8$    | 65.77 | 7.14  | 7.24  | 11.79 | 20.60 | 24.94 | 22.85 | 22.32 | 18.50 | 16.27 |
> | Product-$(\mathbb{H}_\mathbb{R}^8)^4$    | 65.42 | 6.28  | 6.81  | 11.43 | 19.03 | 23.88 | 24.99 | 20.50 | 18.99 | 16.45 |
> | UnitBall-$\mathbb{H}_\mathbb{C}^{16}$    | **84.72** | **52.74** | **44.73** | **39.75** | **35.32** | **33.17** | **32.48** | **29.13** | **29.86** | **28.58** |
>
> Recall that each compressed graph-$(m, k)$ consists of $m$ nodes and is aggregated from $k$ random trees on the $m$ nodes. The bigger $k$ corresponds to the denser and noisier graph. We also give the $\delta$-hyperbolicty of the graphs in the table. When $k=1$ ($\delta=0$), the graph is exactly a tree structure, UnitBall and the product hyperbolic embeddings both have much better performances in this case. When $k>1$, UnitBall still outperforms the product hyperbolic embeddings by a large margin. Especially when $k=2, 3$, the $\delta$ is big, which means the graph deviates from tree structures a lot, the product hyperbolic embeddings fail to reconstruct the graph while UnitBall successfully handles the noisy structures. We add the experiment in Appendix F.11 in our updated revision, where you can check for more results.
>
> **References**:
>
> A. Gu, F. Sala, B. Gunel, and C. Re ́. Learning mixed-curvature representations in product spaces. In ICLR (Poster). OpenReview.net, 2019.
>
> M. Nickel and D. Kiela. Poincare ́ embeddings for learning hierarchical representations. In NIPS, pages 6338–6347, 2017.
>
> M. Nickel and D. Kiela. Learning continuous hierarchies in the lorentz model of hyperbolic geometry. In ICML, volume 80 of Proceedings of Machine Learning Research, pages 3776–3785. PMLR, 2018.
>
> R. Sonthalia and A. C. Gilbert. Tree! I am no tree! I am a low dimensional hyperbolic embedding. In NeurIPS, 2020.

---

> ### Author Response · Authors · 2021-11-16
> **Response to Reviewer PCHz (Part 3: Distance Function and Extension to NNs)**
>
> > Moreover, even after reading the appendix and the experimental section, I do not understand the type of relationship that the distance function in Eq. (9) is supposed to describe. In [B,C,E,F], the motivation of the geodesic distances is given (e.g., high-level nodes tend to be closer to the origin in [B,C]). In the submission, there is no illustration or toy dataset/analysis to understand the embeddings that are learned.
>
> Here we would like to give the illustration about the geometric meaning of the distance function on the unit ball model. The distance function in Eq. (9) maintains the tree-like metric properties. When the points are very close to the origin, it approximates to the Euclidean distance. Additionally, when a point is closer to the origin, it has relatively smaller distances to the other points. Correspondingly, the points near the boundary have very large distances from each other. Therefore, in ideal conditions, the root node of a tree is embedded in the origin while the deeper nodes are embedded farther away from the origin. Recall that the distance function in the real hyperbolic space (Nickel and Kiela, 2017) has similar properties since the real hyperbolic space, as a totally geodesic subspace of the complex hyperbolic space, inherits the tree-like metrics. More details about the Bergman metric and distance function can be referred to Chapter 3.1 in (Goldman, 1999). We incorporate the above illustration in Appendix A.2 in our updated revision.
>
> > Another weakness of the paper is the fact that the learned representations are nonparametric embeddings. For instance, hyperbolic embeddings have been extended to hyperbolic (graph) neural networks, just like ref [E] was extended to graph neural networks [G]. How easy is it to extend the exploited manifold to (graph) neural network and perform inference of test examples?
>
> We would like to clarify the misunderstanding on 'nonparametric embeddings'. Both the optimization-based embedding methods (Nickel and Kiela  (2017;  2018) and our work) and the construction-based embedding methods (Gu et al., 2019; Sonthalia and Gilbert, 2020)  learn the embedding vectors of all the nodes during training. Then in the inference step, the test nodes must have occurred in the training set. For the graph reconstruction task (the tasks focused by the construction-based methods), the test set is the same as the training set, while for the link prediction task, the test queries (i.e., test edges) cannot be leaked in training edges but the test nodes are contained in the training set. On the GNN tasks, e.g., on the experiments of PROD-GCN (Bachmannet al., 2020) which you mentioned in the review, the inference of the test examples also requires the test nodes occurring in the training set.
>
> As for the extension to (graph) neural network, the hyperboloid embeddings (Nickel and Kiela, 2018) has been extended to HGNNs (Liu et al., 2019; Chami et al., 2019) while the Poincare ́ embeddings (Nickel and Kiela, 2017) has been extended to HNNs (Ganea et al., 2018). Our work follows the optimization-based embedding framework and exploits a more powerful geometry. The extension of complex hyperbolic geometry to neural networks can utilize the successful experiences of the abovementioned HGNNs and HNNs. However, there are challenges in carefully formulating the network layers with the operations (e.g., addition, multiplication, exp/log map, etc.) in complex hyperbolic geometry, especially the adaptation to the complex domain. Motivated by our theoretical grounding and empirical success in the shallow embeddings, we believe future work on complex hyperbolic neural networks has a big potential, and our work can inspire explorations on the exciting and challenging problems.
>
> We would like to thank you again for your efforts and time. Please let us know if you have any further concerns or questions about our paper. We look forward to having more discussions with you.
>
> **References**:
>
> I. Chami, Z. Ying, C. R ́e, and J. Leskovec.  Hyperbolic graph convolutional neural networks.  In NeurIPS, pages 4869–4880, 2019.
>
> O. Ganea, G. B ́ecigneul, and T. Hofmann. Hyperbolic neural networks. In NeurIPS, pages 5350–5360,2018.
>
> W. M. Goldman. Complex hyperbolic geometry. Oxford University Press, 1999.
>
> A. Gu, F. Sala, B. Gunel, and C. Re ́. Learning mixed-curvature representations in product spaces. In ICLR (Poster). OpenReview.net, 2019.
>
> Q. Liu, M. Nickel, and D. Kiela. Hyperbolic graph neural networks. In NeurIPS, pages 8228–8239, 2019.
>
> M. Nickel and D. Kiela. Poincare ́ embeddings for learning hierarchical representations. In NIPS, pages 6338–6347, 2017.
>
> M. Nickel and D. Kiela. Learning continuous hierarchies in the lorentz model of hyperbolic geometry. In ICML, volume 80 of Proceedings of Machine Learning Research, pages 3776–3785. PMLR, 2018.
>
> R. Sonthalia and A. C. Gilbert. Tree! I am no tree! I am a low dimensional hyperbolic embedding. In NeurIPS, 2020.

---

### Official Review · Reviewer_1y6j · 2021-11-02

**Correctness:** 3
**Technical Novelty And Significance:** 3
**Empirical Novelty And Significance:** 3
**Recommendation:** 5
**Confidence:** 3

**Main Review:**

- Hyperbolic spaces have gained a lot of interest for representing hierarchical data due to their relative simplicity and computational tractability – both in terms of computing the embeddings and for designing algorithms for downstream applications in such spaces (see, e.g., (Ganea et al., NeurIPS ‘18), (Cho et al., AISTATS ’19), (Weber et al., NeurIPS ‘20)). What implications has the switch to non-constant negative curvature? In comparison with embeddings spaces that are products of constant-curvature manifolds or that don’t have a constraint on the sign of curvature, how simple and tractable are the respective tools? Does this merit the additional representation power that we gain compared to the commonly used models of hyperbolic space?
- How scalable is your approach, especially in comparison to the established hyperbolic embedding approaches you compare against? Some of the real-world data sets that you analyze are larger in size, but I did not see a comment on the computational cost of your approach.
- In your experiments you only compare against two hyperbolic embedding approaches (Nickel and Kiela, NeurIPS ‘17 + ICML ‘18), but not against more recent methods. In particular, it is known that when learning representations in the Poincare model, data points get often mapped to points close to the boundary, which impacts the downstream performance. This can be mitigated by regularizations. It would also be good to include a comparison with embeddings into products of constant-curvature manifolds (e.g., Gu et al., ICRL ‘19).


**Summary Of The Paper:**

The authors propose a new approach for representation learning on hierarchical data. They propose to embed hierarchical data not into the commonly used models of hyperbolic space (the Poincare and Lorentz models), but into complex hyperbolic space instead.

**Summary Of The Review:**

The paper proposes an interesting approach for embedding hierarchical data into complex hyperbolic space. By choosing complex hyperbolic space as the target embedding space, the approach tries to gain additional representation power. I have some conceptual concerns and also think that a more comprehensive experimental comparison with existing approaches would improve the paper (see comments above).

---

> ### Author Response · Authors · 2021-11-16
> **Response to Reviewer 1y6j (Part 1: Downstream Applications of Complex Hyperbolic Embeddings and Computational Cost)**
>
> We would like to thank the reviewer for your helpful comments and questions. In the following, we will address your comments.
>
> > Hyperbolic spaces have gained a lot of interest for representing hierarchical data due to their relative simplicity and computational tractability – both in terms of computing the embeddings and for designing algorithms for downstream applications in such spaces (see, e.g., (Ganea et al., NeurIPS ‘18), (Cho et al., AISTATS ’19), (Weber et al., NeurIPS ‘20)). What implications has the switch to non-constant negative curvature?
>
> As you have noticed, the hyperbolic embedding methods inspired the hyperbolic neural networks for more downstream tasks. For example, the hyperboloid embeddings (Nickel and Kiela, 2018) has been extended to HGNNs (Liu et al., 2019; Chami et al., 2019) while the Poincare ́ embeddings (Nickel and Kiela, 2017) has been extended to HNNs (Ganea et al., 2018). The advantages of complex hyperbolic geometry in shallow embeddings are theoretically analyzed and empirically demonstrated in our paper. The extension of complex hyperbolic geometry to neural networks can utilize the successful experiences of the abovementioned HGNNs and HNNs. However, there are challenges in carefully formulating the network layers with the operations (e.g., addition, multiplication, exp/log map, etc.) in complex hyperbolic geometry, especially the adaptation to the complex domain. Motivated by our theoretical grounding and empirical success in the shallow embeddings, we believe future work on complex hyperbolic neural networks for more downstream applications has a big potential, and our work can inspire explorations on the exciting and challenging problems.
>
> > In comparison with embeddings spaces that are products of constant-curvature manifolds or that don’t have a constraint on the sign of curvature, how simple and tractable are the respective tools? Does this merit the additional representation power that we gain compared to the commonly used models of hyperbolic space?
>
> In comparison with the product space embeddings (Gu et al., 2019), our method does not need to search for the optimal manifold combination among enormous combinations for each new graph. In comparison with the trainable curvature embeddings (Chami et al., 2020), our method does not need to train the extra curvature parameters. Moreover, the trainable curvature method AttH (Chami et al., 2020) learns a curvature parameter for each relation, which makes the trainable curvature not functioning in our single-relation inference task. We compare with AttH in Appendix F.9.
>
> > How scalable is your approach, especially in comparison to the established hyperbolic embedding approaches you compare against? Some of the real-world data sets that you analyze are larger in size, but I did not see a comment on the computational cost of your approach.
>
> As you mentioned in your first comment, representation learning in hyperbolic space has relative simplicity and computational tractability. Our method follows the well-developed optimization-based embedding framework while exploiting a more powerful geometry. In comparison with Poincare ́ embeddings (Nickel and Kiela, 2017) and hyperboloid embeddings (Nickel and Kiela, 2018), the number of updating parameters is the same since we deduct the dimension of UnitBall by half of the baselines. The computations of metrics and distance functions as well as some steps involving complex gradient in RSGD are different, but the complex hyperbolic geometry does not bring more complexities. The computational cost, including time cost and space cost, has little difference with the optimization-based real hyperbolic embeddings (Nickel and Kiela, 2017; 2018).
>
> **References**:
>
> I. Chami, Z. Ying, C. R ́e, and J. Leskovec. Hyperbolic graph convolutional neural networks. In NeurIPS, pages 4869–4880, 2019.
>
> I. Chami, A. Wolf, D. Juan, F. Sala, S. Ravi, and C. Re ́. Low-dimensional hyperbolic knowledge graph embeddings. In ACL, pages 6901–6914. Association for Computational Linguistics, 2020.
>
> O. Ganea, G. B ́ecigneul, and T. Hofmann. Hyperbolic neural networks. In NeurIPS, pages 5350–5360,2018.
>
> A. Gu, F. Sala, B. Gunel, and C. Re ́. Learning mixed-curvature representations in product spaces. In ICLR (Poster). OpenReview.net, 2019.
>
> Q. Liu, M. Nickel, and D. Kiela. Hyperbolic graph neural networks. In NeurIPS, pages 8228–8239, 2019.
>
> M. Nickel and D. Kiela. Poincare ́ embeddings for learning hierarchical representations. In NIPS, pages 6338–6347, 2017.
>
> M. Nickel and D. Kiela. Learning continuous hierarchies in the lorentz model of hyperbolic geometry. In ICML, volume 80 of Proceedings of Machine Learning Research, pages 3776–3785. PMLR, 2018.

---

> ### Author Response · Authors · 2021-11-16
> **Response to Reviewer 1y6j (Part 2: Baselines and Results on Comparison with Product Embeddings)**
>
> > In your experiments you only compare against two hyperbolic embedding approaches (Nickel and Kiela, NeurIPS ‘17 + ICML ‘18), but not against more recent methods.
>
> Please note that except for the Poincar ́e embeddings (Nickel and Kiela, 2017) and hyperboloid embeddings (Nickel and Kiela, 2018), our baseline also includes the SOTA construction-based hyperbolic method TreeRep (Sonthalia and Gilbert, 2020). The performances of TreeRep are thoroughly analyzed and discussed in Section 5.2 \& 5.3 and Appendix F.5 \& F.6.
>
> > In particular, it is known that when learning representations in the Poincare model, data points get often mapped to points close to the boundary, which impacts the downstream performance. This can be mitigated by regularizations.
>
> We agree that the Poincar ́e embedding method has this problem and it can influence its performances such as causing numerical instabilities. We believe the regularization would be a promising solution for this limitation you mentioned.
>
> > It would also be good to include a comparison with embeddings into products of constant-curvature manifolds (e.g., Gu et al., ICRL ‘19).}
>
> As suggested by Reviewer N6rj, although it is impractical to search for the best manifold combination among enormous combinations of product spaces for each new structure, it is worth exploring the comparisons between the complex hyperbolic embeddings and products of hyperbolic embeddings. Therefore, we conduct experiments on the reconstruction task in synthetic compressed graphs. We evaluate the $16$-dimensional UnitBall complex hyperbolic embeddings and $32$-dimensional product hyperbolic embeddings (Gu et al., 2019) on $(\mathbb{H}_\mathbb{R}^2)^{16}$, $(\mathbb{H}_\mathbb{R}^4)^8$, $(\mathbb{H}_\mathbb{R}^8)^4$. The MAP results are reported in the following table.
>
> |  k ( m=500)                              | 1     | 2     | 3     | 4     | 5     | 6     | 7     | 8     | 9     | 10    |
> |------------------------------------------|-------|-------|-------|-------|-------|-------|-------|-------|-------|-------|
> | $\delta$-hyperbolicity                   | 0.0   | 2.5   | 1.5   | 1.0   | 1.0   | 1.0   | 1.0   | 1.0   | 1.0   | 1.0   |
> | Product-$(\mathbb{H}_\mathbb{R}^2)^{16}$ | 66.06 | 7.60  | 7.55  | 11.09 | 20.81 | 24.56 | 24.07 | 21.62 | 18.79 | 16.16 |
> | Product-$(\mathbb{H}_\mathbb{R}^4)^8$    | 65.77 | 7.14  | 7.24  | 11.79 | 20.60 | 24.94 | 22.85 | 22.32 | 18.50 | 16.27 |
> | Product-$(\mathbb{H}_\mathbb{R}^8)^4$    | 65.42 | 6.28  | 6.81  | 11.43 | 19.03 | 23.88 | 24.99 | 20.50 | 18.99 | 16.45 |
> | UnitBall-$\mathbb{H}_\mathbb{C}^{16}$    | **84.72** | **52.74** | **44.73** | **39.75** | **35.32** | **33.17** | **32.48** | **29.13** | **29.86** | **28.58** |
>
> Recall that each compressed graph-$(m, k)$ consists of $m$ nodes and is aggregated from $k$ random trees on the $m$ nodes. The bigger $k$ corresponds to the denser and noisier graph. We also give the $\delta$-hyperbolicty of the graphs in the table. When $k=1$ ($\delta=0$), the graph is exactly a tree structure, UnitBall and the product hyperbolic embeddings both have much better performances in this case. When $k>1$, UnitBall still outperforms the product hyperbolic embeddings by a large margin. Especially when $k=2, 3$, the $\delta$ is big, which means the graph deviates from tree structures a lot, the product hyperbolic embeddings fail to reconstruct the graph while UnitBall successfully handles the noisy structures. We add the experiment in Appendix F.11 in our updated revision, where you can check for more results.
>
> We would like to thank you again for your efforts and time. Please let us know if you have any further concerns or questions regarding our paper. We look forward to having more discussions with you.
>
> **References**:
>
> A. Gu, F. Sala, B. Gunel, and C. Re ́. Learning mixed-curvature representations in product spaces. In ICLR (Poster). OpenReview.net, 2019.
>
> M. Nickel and D. Kiela. Poincare ́ embeddings for learning hierarchical representations. In NIPS, pages 6338–6347, 2017.
>
> M. Nickel and D. Kiela. Learning continuous hierarchies in the lorentz model of hyperbolic geometry. In ICML, volume 80 of Proceedings of Machine Learning Research, pages 3776–3785. PMLR, 2018.
>
> R. Sonthalia and A. C. Gilbert. Tree! I am no tree! I am a low dimensional hyperbolic embedding. In NeurIPS, 2020.

---

> ### Author Response · Authors · 2021-11-22
> **Thanks for your reply**
>
> Thanks for letting us know that your concern regarding the motivation of complex hyperbolic geometry remains. We would like to explain the motivation again for a clearer understanding. The real hyperbolic space resembles tree metrics and represents tree structures naturally because of the constant negative curvature. As a result, the hyperbolic embeddings can improve over the Euclidean embeddings in hierarchical structures, but it still suffers from limitations since most real-world hierarchically structured data such as taxonomies and multitree networks have varying local structures and they are not trees, thus they **do not ubiquitously match the constant curvature property of the hyperbolic space**. In comparison, **complex hyperbolic geometry has a more powerful representation capacity for the general hierarchical structures** because of its favorable geometric properties. Specifically, **the variable negative curvature of the complex hyperbolic geometry can address the abovementioned limitation of hyperbolic embeddings**. In addition, **the complex hyperbolic geometry still maintains the tree-like exponential volume growth property**. The geometric properties are introduced in Section 3 and the empirical evaluations in Experiments demonstrate the advantages of complex hyperbolic embeddings.
>
> Please also kindly note that in our previous responses, we addressed all your comments, including:
> * Addressed your concerns about our tractability advantages compared with products of constant-curvature manifolds and trainable curvature embeddings.
> * Responded to your question about scalability.
> * Pointed out that we have the SOTA construction-based hyperbolic method TreeRep as our baseline.
> * Reported the supplemented experimental results on comparison with product embeddings.
>
> We sincerely hope our response can clarify any possible confusion. We appreciate your help to improve our paper very much. More specific comments or suggestions are favorably received if you find any other problems in our work.

---

### Author Response · Authors · 2021-11-16
**Many thanks to all reviewers and Summary of the revisions**

We sincerely appreciate the comments and feedback given by the reviewers. The suggestions and comments help us to improve our paper. Here we summarize the revisions in our updated version.

1. We conduct experiments on comparison with the product hyperbolic embeddings and report the results in Appendix F.11.

2. We conduct experiments to find the nodes whose embeddings lie in the totally real plane of the unit ball model and present the result in Appendix F.12.

3. We add related works on other research tasks and applications inspired by hyperbolic learning in Section 2.

4. We incorporate the illustration about the distance function on the unit ball model in Appendix A.2.

We hope our responses and the updated revision can address your concerns and clear your confusion. We are delighted to have further discussions.

---

> ### Author Response · Authors · 2021-11-22
> **A new revision uploaded**
>
> Hi all reviewers, we uploaded a new revision. Besides the previous four changes, we added more experimental results on comparison with the product hyperbolic embeddings in Appendix F.11. Thank you for your attention and time!

---

### Decision · Program_Chairs · 2022-01-20

**Decision:**

Reject

**Comment:**

The paper proposes to learn embeddings into complex hyperbolic space. This is an extension of the popular hyperbolic-space embeddings which have shown success on graph-like and tree-like data. Reviews and discussion mostly centered around the lack of clear motivation for the work (why complex hyperbolic spaces?) and the lack of a clear advantage over other manifold embedding methods that have varying curvature. The reviewers mentioned many questions and points that they thought the work should cover. There was also concern about the baselines against which the method was compared. There was not a consensus that the paper should be accepted, and no reviewer argued strongly for acceptance, even after the author response. As a result, I recommend that this paper not be accepted at this time. I expect a new version of this paper, incorporating this reviewer feedback and especially improving the explanation of the motivation, will be a good submission for a future conference.